# Sensing Systems for Respiration Monitoring: A Technical Systematic Review

**DOI:** 10.3390/s20185446

**Published:** 2020-09-22

**Authors:** Erik Vanegas, Raul Igual, Inmaculada Plaza

**Affiliations:** EduQTech, Electrical/Electronics Engineering and Communications Department, EUP Teruel, Universidad de Zaragoza, 44003 Teruel, Spain; 790974@unizar.es (E.V.); inmap@unizar.es (I.P.)

**Keywords:** respiratory monitoring, respiration sensor, breathing sensor, sensor comparison, systematic review, comprehensive review, technical review

## Abstract

Respiratory monitoring is essential in sleep studies, sport training, patient monitoring, or health at work, among other applications. This paper presents a comprehensive systematic review of respiration sensing systems. After several systematic searches in scientific repositories, the 198 most relevant papers in this field were analyzed in detail. Different items were examined: sensing technique and sensor, respiration parameter, sensor location and size, general system setup, communication protocol, processing station, energy autonomy and power consumption, sensor validation, processing algorithm, performance evaluation, and analysis software. As a result, several trends and the remaining research challenges of respiration sensors were identified. Long-term evaluations and usability tests should be performed. Researchers designed custom experiments to validate the sensing systems, making it difficult to compare results. Therefore, another challenge is to have a common validation framework to fairly compare sensor performance. The implementation of energy-saving strategies, the incorporation of energy harvesting techniques, the calculation of volume parameters of breathing, or the effective integration of respiration sensors into clothing are other remaining research efforts. Addressing these and other challenges outlined in the paper is a required step to obtain a feasible, robust, affordable, and unobtrusive respiration sensing system.

## 1. Introduction

Continuous monitoring of physiological variables is essential for health and well-being applications. One of the most interesting physiological variables is respiration. Breathing information is useful for health condition assessment [1]. It can help diagnose respiratory diseases, such as asthma, sleep apnea, and chronic obstructive pulmonary diseases (chronic bronchitis, emphysema, and non-reversible asthma) [2]. It is also used to identify heart failure or heart attack [3] and may serve as an indicator of changes in the nervous system, cardiovascular system, or excretory system, among others [4]. Once a disease has been diagnosed, breathing monitoring may be used during the treatment or for the surveillance of patients. It also plays a relevant role in the monitoring of newborn babies. Some of them are born under delicate conditions, and this monitoring may avoid any casualty due to infant sleep apnea [5]. Older people suffering from age-related conditions and diseases, like Parkinson or dementia [6], and sedentary patients could also benefit from unobtrusive health surveillance [7].

Breathing monitoring is also applicable to the field of work health and safety at work [8]. Firstly, having breathing information from workers can be helpful in assessing work-related risks to plan preventive actions to be undertaken before a work disease appears. The analysis of respiratory information may lead to the design of safer work places. Secondly, respiratory monitoring may help prevent job accidents and is especially useful for jobs, such as plane piloting, industrial machine drivers, car, bus, or train drivers, who can benefit from having breathing information on real time [9]. 

Respiratory monitoring has also been applied in the analysis of human emotions [10,11]. Respiratory rate (RR) can be associated with emotions, such as fear, stress, anger, happiness, sadness, or surprise [12]. This can be used to prevent mental diseases and in the treatment of patients with mental disorders. Human emotions are also useful in psychological studies, for example, to assess or understand consumer and social trends [13]. They have also been applied in assessing the level of safety of drivers [14] by monitoring their emotional state. They were also used in the computer science field to improve software engineering processes, overcoming the limitations of usability questionnaires and helping to provide more personalized web experiences. For example, they can be used to obtain information about consumer behavior on websites and their interactions. Respiratory monitoring may contribute to real time recognition of emotions, which is an area of active research in the video game industry to generate dynamic gaming experiences [15]. There are also applications in the education field and e-Learning. Some emotional states have positive effects on learning processes, while others hinder them. It is possible to personalize the learning process by providing the most effective resources for each emotional state [16].

Respiratory information is also applied in the sports field to monitor the performance of athletes during their activities [17,18]. This information can be used to optimize their training or to prevent health problems. Similarly, it is used in Magnetic Resonance Imaging (MRI) machines to guarantee the good conditions of patients throughout the process [19] and to reduce their level of stress [20].

Another less common application of respiratory sensing is the evaluation of the health of combat soldiers [21,22]. This has a double utility: it provides information on the integrity of soldiers and allows collecting field information. Breathing monitoring has also been used for emergency situations, such as rescue of or searching for people, in which breathing information is required in a non-contact way for faster and more effective intervention [23]. 

Figure 1 shows an overview of the applications of respiration monitoring.

To perform respiratory monitoring, several approaches were proposed in the literature [24]. Monitoring systems use sensors to measure breathing parameters. There are large differences among approaches depending on sensing techniques and sensors, breathing parameters, sensor locations, system setups, communication protocols, processing stations, energy autonomy and power consumption, field of application, algorithms used to process sensor data, software of analysis, and performance evaluation, among others. Given that the number of studies and approaches has increased dramatically in recent years, it may be useful to review existing systems, discussing trends, challenges and issues in this field.

There are several existing reviews in the field of wearable sensors. For example, the survey of Mukhopadhyay et al. [25] focused on wearable sensors to monitor human activity continuously. They described the typical architecture of a human activity monitoring system based on sensors, microcontrollers, communication modules, and remote processing. The paper outlined transmission technologies and energy harvesting issues and predicted an increase in interest in wearable devices in the near future. Similarly, the work of Nag et al. [26] reviewed flexible wearable sensors to monitor physiological parameters. The study focused on the materials used to manufacture sensors based on different factors, such as application, material availability, cost, or manufacturing techniques. Different operating principles were identified: electromechanical, pressure and strain, chemical, and magnetic field-based, among others. The transmission technologies used in the sensing systems and their possible applications were also reviewed in detail. Finally, the paper identified several challenges and future opportunities. The most relevant was the expected reduction in the cost of manufacturing flexible sensing systems. However, this paper focused exclusively on flexible sensing systems, and no review of other technologies was performed. In addition, it did not specifically address respiration sensing, but instead considered sensors for any type of physiological parameter. Similarly, the reviews of Chung et al. [27] and Bandodkar et al. [28] also focused on wearable flexible sensors, but specifically targeted at sweat analysis. Meanwhile, the review of Lopez-Nava et al. [29] addressed inertial sensors for human motion analysis. Different aspects were studied: sensor type, number of sensing devices and their combination, processing algorithms, measured motion units, systems used for comparison, and number of test subjects and their age range, among others. The review identified a trend toward low-cost wearable systems. 

Seshadri et al. [30] presented a work focused on wearable sensors to monitor athletes’ internal and external workload. The paper addressed wearable devices for athletes comprehensively, including physical performance, physiological and mental status, and biochemical composition. RR was considered as one more physiological parameter. In fact, sensors to measure position, motion, impact, biomechanical forces, heart rate, muscle oxygen saturation, and sleep quality were also considered. The paper concluded that wearable sensors had the potential to minimize the onset of injuries and evaluate athlete performance in real time. 

Aroganam et al. [31] reviewed wearable sensors for sport applications excluding professional sports. Communication technologies, battery life, and applications were widely discussed. The paper concluded that inertial and Global Positioning System (GPS) sensors were predominant in sport wearables. A gap was detected in user experience studies of existing devices. Meanwhile, Al-Eidan et al. [32] presented a systematic review on wrist-worn wearable sensors. They focused on user interface, interaction, and use studies of the sensing systems. Processing techniques were also analyzed showing high variability among them and including machine learning techniques and threshold-based methods. Similarly, validation experiments lasted from 2 s to 14 weeks and most of the experiments were performed under laboratory conditions. Few studies presented real-world setups with target users. Other aspects analyzed were sampling frequencies and features extracted. Challenges of wrist-based systems were identified in relation to weight, battery life, lack of standardization, safety, user acceptance, or design. 

Mansoor et al. [33] performed a review on wearable sensors for older adults. The review focused on sensor target population, sensor type, application area, data processing, and usability. Fourteen papers were analyzed. They identified barriers, such as inaccurate sensors, battery issues, restriction of movements, lack of interoperability, and low usability. The paper concluded that these technical challenges should be resolved for successful use of wearable devices. 

Heikenfeld et al. [34] conducted a review on wearable sensors that interfaced with the epidermis. Wearable sensors were classified into four broad groups: mechanical, electrical, optical, and chemical. Several subgroups were identified within each category. Body-to-signal transduction, actual devices and demonstrations, and unmet challenges were discussed. The paper concluded that, in general, sensing categories had remained isolated from each other in commercial products, and strategies were still needed to easily attach and detach disposable systems.

Witte and Zarnekow [35] reviewed wearable sensors for medical applications. Ninety-seven papers were analyzed in relation to disease treatments, fields of application, vital parameters measured, and target patients. The paper identified a trend toward heart and mental diseases monitoring. Sensors were used for monitoring or diagnosis, collecting physical activity data, or heart rate data. The work of Pantelopoulos et al. [36] surveyed wearable biosensor systems for health monitoring. The design of multiparameter physiological sensing systems was discussed in detail. Meanwhile, the study of Liang et al. [37] addressed wearable mobile medical monitoring systems. Emphasis was placed on devices based on wireless sensing networks, and special attention was given to textile technologies. Finally, the paper of Charlton et al. [38] reviewed the estimation of the RR using two different signals: the electrocardiogram (ECG) and the pulse oximetry (photoplethysmogram, PPG).

A recent review on contact-based sensors to measure RR was published by Massaroni et al. [24]. This paper identified seven contact-based techniques: measuring of respiratory airflow, respiratory sounds, air temperature, air humidity, air components, chest wall movements, and modulation cardiac activity. Several possible sensors could be used for each technique. Some of the sensors identified in the review were flowmeters, anemometers, fiber optic sensors, microphones, thermistors, thermocouples, pyroelectric sensors, capacitive sensors, resistive sensors, nanocrystal and nanoparticles sensors, infrared, inductive, transthoracic, inertial, ECG sensors, and PPG sensors, among others. The paper presented a detailed description of each sensing technology, focusing on metrological properties and operating principles. Equations were provided for most sensors. In addition, the study compared the optimal techniques for clinical settings (respiratory airflow, air temperature, air components, chest wall movements, and modulation of cardiac activities), occupational settings (respiratory airflow, air components, and chest wall movements) and sport and exercise (respiratory airflow and chest wall movements). These techniques were considered optimal for controlled environments. 

A previous work on respiration sensors was published by AL-Khalidi et al. [39]. This paper covered both non-contact and contact-based methods and provided a general description of several sensing techniques. On the one hand, contact-based technologies included five sensing methods: acoustic, airflow detection, chest and abdominal movement measuring, transcutaneous CO_2_ monitoring, oximetry prove (SpO_2_), and electrocardiogram derived methods. On the other hand, non-contact technologies included radar-based detection, optical methods, thermal sensors, and thermal imaging. The paper concluded that non-contact RR monitoring had advantages over contact methods since they caused the least discomfort to patients. 

Three other related surveys were published, to the best of our knowledge. The review by van Loon et al. [40] studied respiratory monitoring from a hospital perspective without analyzing technical items. The review of Rajala and Lekkala [41] focused exclusively in the film-type sensor materials polyvinylidenefluoride (PVDF) and electro-mechanical film (EMFi), while the recent review of Massaroni et al. [42] analyzed fiber Bragg grating sensors for cardiorespiratory monitoring. 

In this paper, we present a survey on sensing systems for respiratory monitoring. This paper has several novelties with respect to the existing reviews in the state of the art:This review is not exclusively focused on sensor metrological properties or operating principles. Instead, this paper also reviews all the different aspects involved in the design and development of a respiration sensing system: communication protocols, processing stations, energy autonomy and power consumption, general system setups, sensor location and size, breathing parameters, validation methods, details of the test experiments, processing algorithms, software used for analysis, and performance evaluation. To the best of our knowledge, this is the first review paper that analyzes all these aspects in breathing sensors.This paper does not focus exclusively on RR. In addition, sensors that measure other breathing parameters are also surveyed.Unlike previous reviews, this survey is systematic. Studies on respiration sensors were obtained using objective selection criteria. They were then subjected to detailed analysis.

Therefore, this paper provides a comprehensive overview of all aspects to consider in the design of respiratory sensing systems. It aims to help engineers and researchers to identify the different options at each design stage. 

The structure of this review is as follows: Section 2 presents the study design, selection criteria, and organization of the review results; Section 3 describes the results of the literature search, which are classified into different groups, the items of analysis and the results of the analysis of those items for each study; Section 4 discusses the trends in respiratory monitoring, the issues in the design of respiration sensors, and the current challenges in this field, highlighting the research opportunities; and, finally, Section 5 draws some conclusions.

## 2. Materials and Methods

### 2.1. Search and Selection Procedure

A systematic search of the literature was carried out to identify relevant papers in the field of sensors for respiratory monitoring. The IEEE (Institute of Electrical and Electronics Engineers) Explore and Google Scholar were used for this review. IEEE Explore is a reference in engineering studies and Google Scholar provides a broader perspective to complement the results. Four sets of keywords were selected to perform the searches. To identify these keywords, a preliminary study was conducted that examined key studies in this field. As a result, the five search terms selected were the following: (1) “breathing” plus “monitoring”, (2) “respiratory” plus “monitoring”, (3) “breathing” plus “sensor”, (4) “respiratory” plus “sensor”, and (5) “respiration” plus “sensor”.

To analyze the most recent research, articles from 2010 to 2019 were considered. Searches were conducted in February 2019 and repeated in March 2020. The sort by relevance of IEEE Xplore and Google Scholar was used to obtain the most relevant articles first. According to the official IEEE Xplore website, the search results are “sorted by how well they match the search query as determined by IEEE Xplore” [43]. Regarding the relevance criteria of Google Scholar, its official website points out that the rank is made by “weighting the full text of each document, where it was published, who it was written by, as well as how often and how recently it has been cited in other scholarly literature” [44]. Journals, magazines, and conferences were considered in the searches. As a result of the five searches in the two repositories, more than a million results were obtained. For each search and repository, the 100 most relevant papers were selected, resulting in 1000 studies. This number is high enough to provide a comprehensive review of the topic. The title and abstract of all these studies were examined and those not related to the subject of the review were discarded, resulting in 236 papers. Then, a second selection was made based on the content of the papers, discarding those that did not deal with sensors for respiratory monitoring. Finally, 198 papers were obtained. All of them were subjected to a detailed analysis that is presented in Section 3. Figure 2 (top) shows an overview of the selection procedure. Figure 2 (bottom) presents the PRISMA diagram that details the item selection process [45].

### 2.2. Organization of the Results

The search results were analyzed in detail. For that, papers were divided into two categories: wearable systems and environmental systems. This is a typical classification found in several sensor-related studies [24,46]. Wearable methods require individuals to carry the sensors, while environmental methods place them around subjects. The wearable category includes 113 studies, while the environmental category comprises the remaining 85 studies.

Different aspects of respiratory sensing systems were analyzed for each paper. The items selected can be divided into four categories (Figure 3): (1) sensor and breathing parameter, (2) data transmission and power consumption, (3) experiments performed for sensor validation, and (4) sensor measurement processing.

The category “sensor and breathing parameter” includes the following items of analysis: (1.1) sensing technique and sensor, (1.2) breathing parameter, and (1.3) sensor location and size. 

Four items are included in the category “data transmission and power consumption”: (2.1) general system setup, (2.2) communication protocol, (2.3) processing station, and (2.4) energy autonomy and power consumption.

The category “sensor validation” comprises several items related to the design of experiments to validate the sensors (they are listed in Section 3.3). 

Three items are included in the “sensor measurement processing” category: (4.1) performance evaluation, (4.2) processing algorithm, and (4.3) software used for the analysis.

For each category, we first describe in detail the different items of analysis, except item (1.1) “sensing technique and sensor”, which was described extensively in the review of Massaroni et al. [24]. Then, we provide the value of those items for each study selected for both categories (wearable and environmental). Results were subjected to critical analysis and discussion.

## 3. Results

This section has been structured around the four categories of analysis introduced in Section 2.2. First, the items of analysis and their possible values are described in detail for each category (Section 3.1.1, Section 3.2.1, Section 3.3.1 and Section 3.4.1). Then, the values of those items provided in the studies selected are analyzed and discussed (Section 3.1.2, Section 3.2.2, Section 3.3.2 and Section 3.4.2).

### 3.1. Sensor and Breathing Parameter

#### 3.1.1. Items of Analysis

This category includes the following items of analysis: sensing technique and sensor, breathing parameter, and sensor location and size. 

##### Sensing Technique and Sensor

According to the review of Massaroni et al. [24], two different dimensions can be observed in the operating principle: the technique selected to obtain respiration information and the sensor used to capture that information. For each possible technique, there are several sensors available. 

To classify the papers analyzed in this review, the classification established in the work of Massaroni et al. [24] was used. It was expanded to also cover environmental breathing sensors. The techniques and sensors identified were:Technique based on measurements of respiratory airflow. Possible sensors are differential flowmeters, turbine flowmeters, hot wire anemometers, photoelectric sensors, and fiber optic sensors.Technique based on measurements of respiratory sounds. Possible sensors are microphones.Technique based on measurements of air temperature. Possible sensors are thermistors, thermocouples, pyroelectric sensors, fiber optic sensors, infrared sensors, and cameras.Technique based on measurements of air humidity. Possible sensors are capacitive sensors, resistive sensors, nanocrystal and nanoparticles sensors, impedance sensors, and fiber optic sensors.Technique based on measurements of chest wall movements. Three different types of measurement were identified in this technique:
○Strain measurements: Possible sensors are resistive sensors, capacitive sensors, inductive sensor, fiber optic sensors, piezoelectric sensors, pyroelectric sensors, and triboelectric nanogenerator.○Impedance measurements: Possible sensors are transthoracic impedance sensors.○Movement measurements: Possible sensors are accelerometers, gyroscopes and magnetometers, frequency shift sensors, DC (direct current) generators, ultrasonic proximity sensors, cameras, optical sensors, inductive sensors, and Kinect sensors. Technique based on measurements of modulation cardiac activity. Possible sensors are ECG sensors (for biopotential measurements), PPG sensors (for light intensity measurements), radar sensors, and Wi-Fi transmitters and receivers.

Equations and details of the different sensors are included in the reference review paper [24].

##### Breathing Parameters

Breathing parameters are the metrics provided as output of the sensing process. Possible breathing parameters are the following:*Respiratory rate (RR)*: Number of breaths (inspiration and expiration cycles) performed by a subject in one minute (Figure 4). It is measured in breaths/min (bpm). Other metrics derived from the RR can also be calculated [10]:
○Breathing period: Time duration of a breathing cycle(s).○Inspiratory time: Part of the breathing period that corresponds to inspiration (s). According to Figure 4A, it can be obtained as $t_{b}-t_{a}$○Expiratory time: Part of the breathing period that corresponds to expiration (s). According to Figure 4A, it can be obtained as $t_{d}-t_{c}$

*Volume parameters*: Metrics that provide volume information obtained from inhaled or exhaled air during breathing. Volume metrics comprise a set of sub-metrics related to the volume of air available in the lungs [47]. Some of the metrics that could be found in the breathing sensor studies were:
○*Tidal volume (TV)*: It is the volume of air inhaled or exhaled during normal respiration (without forcing breathing). It is measured in liters (L). From the volume versus time signal represented in Figure 4B, the TV for a given breathing period could be calculated as TV=|Vn−1−Vn|=|Vn−Vn−1|, where *V_n_* is the air volume associated with the *n* respiration peak or valley. ○*Minute volume (MV)*: It is the volume of air inhaled or exhaled by a subject in one minute during normal breathing. It is measured in L/min. It can be roughly obtained from the TV and the RR as MV=TV·RR. From the representation of Figure 4B, the MV can be calculated as MV=∑i=2n|Vi−1−Vi|; ∀i∈ℤ:i∈[2,n]:2|i, where *n* is the number of peaks (or valleys) in the air volume curve that can be found in one minute of breathing.○*Peak inspiration flow (PIF)*: According to Warner and Patel [47], it is the maximum flow at which a given tidal volume breath can be delivered. It is measured in L/min. From the representation of Figure 4B, it can be obtained as PIF=(Vb−Va)/(tb−ta), where *(V_a_, t_a_)* is the point associated with the valley in the time-volume curve before inspiration, and *(V_b_, t_b_)* is the point related to the peak of inspiration at which the given tidal volume is delivered.○*Exhalation flow rate (EFR)*: Volume of air exhaled per time unit. It is expressed in L/s and can be calculated as EFR=(Volume of exhaled air)/(Exhalation time) [48]. From the representation of Figure 4B, it can be obtained as EFR=(V3−V4)/(t4−t3), where *(V*_3_*, t*_3_*)* is the point corresponding to a peak of the time-volume curve, and *(V*_4_, *t*_4_*)* is the next valley of the curve.○There are other air volume metrics, such as peak expiratory flow (maximum flow at which a given tidal volume can be exhaled; it can be obtained as (Vc−Vd)/(td−tc) from Figure 4B), vital capacity (volume of air expired after deep inhalation; it can be obtained as Ve−Vf  from Figure 4B), or forced vital capacity (same as vital capacity but maximizing the expiratory effort; it can be obtained as Vg−Vh  from Figure 4B), among others [49]. They have barely been used in breathing sensor studies.○*Compartmental volume*: Instead of considering air volume, this metric measures the change in volume of breathing-related body parts, like chest, thorax, or abdomen [49].

*Respiration patterns*: There are studies in which the purpose is to identify patterns in the signals obtained from the recording of respiration instead of providing a particular breathing parameter. Common patterns identified are abnormal breathing [50,51,52], apnea episodes [50,51,53,54], Kussmaul’s respiration, Cheyne-Stokes breathing, Biot’s respiration, Cheyne-Stokes variant, or dysrhythmic breathing, among others [53]. There are also studies that identified the type of breathing (heavy or shallow breathing, mouth breathing, abdominal breathing, or chest breathing) [53].

##### Sensor Location and Size

Sensor location and size play a relevant role in system usability and can determine the acceptability of the technology by its potential users [55,56]. Figure 5 shows possible locations for wearable systems. The locations are chest (diaphragm or pectoral muscle), abdomen, waist, arm, forearm, finger, mouth (including mouth mask), nose (nasal bridge, above lip or nostril), wrist, neck (suprasternal notch area), or back. Regarding environmental systems, sensors can be located at a fixed distance from the subject, can be integrated into an object commonly used by the subject (pillow, mat, mattress, etc.), or can be distributed on nodes, among others. The location of a sensor largely depends on its operating principle and the specific application.

#### 3.1.2. Results of the Analysis

Table 1 presents the results of the analysis of the items technique, sensor, parameter, and location and size for the studies in the wearable category, while Table 2 shows the results for the environmental papers.

In relation to the sensing techniques and sensors, Figure 6 and Figure 7 show the main results for the wearable and environmental categories, respectively. Most authors chose to detect chest wall movements (60%). For the environmental category, modulation of cardiac activity was also very common [5,7,50,52,54,163,164,168,174,175,177,178,189,190,196,197,198,199,205,206,207,213,214,219,221,223,224,225,226,229,232,233]. Meanwhile, air temperature and air humidity were the second [2,64,91,104,107,116,120,128,133,145,153,154,155,156,162] and third [66,74,75,80,82,89,97,101,137,138] most widely implemented techniques in the wearable category, at great distance. In this category, fiber optic sensors were used in almost 19% of the studies, resistive sensors in 15%, accelerometers in more than 11%, and capacitive sensors in more than 9%. Great variability in sensors can be found in studies of this category, as there is no predominant type. This contrasts with the environmental category since radars are used in more than 33% of the studies, being the leading technology followed by cameras (18%) and fiber optic sensors (14%). There are types of sensors, such as magnetometers, gyroscopes, microphones, optical sensors, inductive sensors, or thermistors, in which its use is very limited in both categories [62,73,76,77,81,91,95,106,107,115,119,120,131,135,156,160,162,185,193,217].

Regarding breathing parameters (Figure 8), RR was obtained in 60% of the wearable studies and in 79% of the environmental studies. It was the most widely used parameter by far. Other metrics based on the analysis of the magnitude versus time curve, such as breathing period or expiratory/inspiratory times, were barely used (2% in the wearable category) [94,103]. The representation of the volume versus time curve or the use of volumetric parameters was not common. They appeared in 10% of the studies of the wearable category [2,17,49,61,67,111,113,116,122,127,147] and in 5% of the studies of the environmental group [48,51,52,215]. Among the possible volume metrics, tidal volume was the most common in the wearable category [2,17,49,61,111,113,116,122,127], while it was found in one study of the environmental category [52]. The rest of the metrics (MV, vital capacity, peak inspiratory flow, peak expiratory flow, and compartmental volume) were used in isolated cases. A considerable number of studies detected respiratory patterns in both wearable [17,143,152,159] and environmental categories [10,19,50,51,52,53,54,180,182,194,218]. The most common approach was to detect abnormal breathing patterns to identify respiratory disorders, such as apnea. This was especially common in environmental systems.

Regarding sensor location, most wearable studies placed them on the chest or abdomen (Figure 9). This was the most common trend by far. It was also common that sensors were embedded in shirts at chest or abdomen level [21,49,59,65,69,84,85,94,108,113,123,142,143,151,235]. This was the location selected by 15% of the studies. Nose or mouth were also widely used locations to place the sensors. As a particular case of sensors placed in the nose or mouth, several researchers integrated them into a mask [66,75,80,82,92,101,107,137,156]. This contrasts with locations, like fingers, waist, arms, or wrists, in which use was residual [93,115,117,118,126,139,157]. 

Figure 10 shows the locations adopted in the environmental studies. On the one hand, the most common approach was to place the sensor at a fixed distance from the subject. Fifty-two% of the studies used this setup. On the other hand, Figure 10 shows that placing the sensors as nodes without precise control of the distance between the sensor and the subject was adopted by 6% of the studies. Meanwhile, 29% of the studies integrated the sensors into mats or pillows [9,19,164,165,166,169,170,173,179,182,183,186,194,201,202,203,210,211,212,217,218,220,227,230,231,236] to measure breathing parameters during rest activities mainly. The rest of the environmental locations shown in Table 2 were only used in isolated cases.

### 3.2. Data Transmission and Power Consumption

#### 3.2.1. Items of Analysis

This category includes the following items of analysis: general system setup, communication protocol, processing station, and energy autonomy and power consumption.

##### General System Setup

Different configurations can be found in systems for respiratory monitoring depending on the data transmission architecture. Systems can be roughly divided into two categories (Figure 11): (A) those that perform data processing on a centralized processing platform and (B) those that perform data processing near the remote sensing unit.

*Systems that perform centralized processing*: Data processing is done in a centralized system that does not need to be close to the subject being monitored. The magnitude values registered by the sensors are acquired and conditioned [24] and then transmitted to a centralized processing unit. Three different approaches can be found depending on the specific point where the acquisition & conditioning module and transmission module are placed:
○The acquisition & conditioning and transmission modules are in the same package as the sensing unit (cases 1.x of Figure 11A, ∀x ∈ [1..2]).○The acquisition & conditioning module is in the same package as the sensing unit, but the transmission module is placed externally (cases 2.x of Figure 11A, ∀x ∈ [1..2]).○Both the acquisition & conditioning and transmission modules are not included in the same package as the sensing unit (cases 3.x of Figure 11A, ∀x ∈ [1..2]).


For all three approaches, data visualization can be done in two different ways: next to the processing unit of the registered signals (cases 1.1, 2.1, and 3.1 of Figure 11A) or at a different point (cases 2.1, 2.2, and 3.2 of Figure 11A).

*Systems that perform remote processing*: Processing of breathing signals to determine the respiratory parameters of interest is performed near the subject whose breathing is being monitored. Three different setups are possible depending on whether the acquisition & conditioning module and the processing module are included in the same package as the sensing unit:
○The acquisition & conditioning circuits, the microcontroller for the processing and the data transmission module are placed in the same package as the sensing unit (cases 4.x of Figure 11B, ∀x ∈ [1..2]). ○The acquisition & conditioning circuits are placed in the same package as the sensing unit. However, the microcontroller in charge of the processing and the data transmission module are placed in an external package, which is not compactly integrated with the sensing module (cases 5.x of Figure 11B, ∀x ∈ [1..2]).○The acquisition & conditioning circuits, the microprocessor and the data transmission module are placed in a different package than the sensing unit (cases 6.x of Figure 11B, ∀x ∈ [1..2]).

Regarding data visualization, it can be done in two different ways: remotely without the need for data transmission (in this case, the data transmission module is not included) (cases 4.1, 5.1, and 6.1 of Figure 11B) or in a central unit (cases 4.2, 5.2, and 6.2 of Figure 11B).

##### Communication Protocol

Communication between the different modules of the system can be classified according to whether it is wired or wireless:*Wired transmission*: All system elements (sensing, acquisition, conditioning, transmission, processing, and visualization) are physically connected. The USB (universal serial bus) protocol is the most common way of transmitting the acquired respiratory signals.*Wireless transmission*: Subjects wear the sensing system without cable connections to other elements of the system. The transmission and reception of measurements is carried out through a wireless transmission technology. Therefore, the usability of the system increases [55]. Different transmission technologies can be found in existing studies [237]:
○*Bluetooth*: It is a standard and communication protocol for personal area networks. It is suitable for applications that require continuous data transmission with a medium data transmission rate (up to 1 Mbps). It uses a radio communication system, which means that the transmitting and receiving devices do not need to be in line of sight. It operates in the 2.4–2.485 GHz band with a low transmission distance (1 to 100 m, typically). There are five Bluetooth classes (1, 1.5, 2, 3, and 4). Most Bluetooth-based respiration monitoring systems use class 2 or higher. This means that the transmission distance is short (less than 10 m, in general), but the power consumption is also moderate [237].○*Wi-Fi*: This technology is generally used for local area networks instead of personal area networks, like Bluetooth. It has much higher data transmission rates and power consumption is also higher. At a typical 2.4 GHz operating frequency, it can consume a maximum of 100 mW. Wi-Fi operating band is in the 2.4–5 GHz range. In general, the transmission range is between 50 m and 100 m, although it can be greatly extended in some conditions. This technology is suitable for applications where constant high-speed data transmission is required, the transmission distance is relatively large, and power consumption is not an issue [238]. ○*GSM/GPRS*: Global System for Mobile Communications (GSM) is a standard for mobile communication that belongs to the second-generation (2G) of digital cellular networks. It requires base stations to which the mobile devices connect. The coverage range of base stations varies from a few meters to dozen of kilometers. Within this 2G technology, it is also possible to find the General Packet Radio Service (GPRS), which is data-oriented. The transmission rate of GPRS is low (around 120 kbps, although this rate is usually lower in real conditions) with a limitation of 2 W of power consumption. The frequency band of this technology is in the range of 850–1900 MHz [239]. ○*Zigbee*: It is a specification of several high-level communication protocols. Zigbee is used for the creation of personal area networks that do not need high data transmission rates. ZigBee can operate in the industrial, medical and scientific radio bands, which may vary among countries. This is the reason why it generally works in the 2.4 GHz band that is available worldwide. If the system operates in the 2.4 GHz band, its data transmission rate is 250 kbps. Devices using this technology are generally inexpensive since the required microprocessor is simple due to the low transmission rate of Zigbee. Power consumption is low since nodes can be asleep until some information is received. It is useful for applications that do not require constant transmission. The range of transmission distance is similar to that of Bluetooth technology [237]. ○*Radio frequency*: These modules are suitable for applications that do not need a high speed of data transmission. Radio frequency works in the Ultra High Frequency band (433 MHz) and requires a receiver-transmitter pair. It is low power and cheap, with a small module size. Communication range is from 20 to 200 m. This range depends on the input voltage of the module: at higher voltages, greater communication distance is reached. Working voltage for this technology ranges from 3.5 to 12 V. Radio broadcasting is performed through amplitude modulation. Radio frequency requires both receivers and transmitters to incorporate a microcontroller module. Typical power consumption is up to 10 mW.

Table 3 shows a schematic comparison of some key properties of the main wireless transmission technologies used in respiration studies.

##### Processing Station

Another item of analysis is the platform on which the recorded signals are processed to obtain respiratory information. Several options exist in the state of the art (Figure 12):PC (personal computer): The respiration sensing system is connected or linked to a local PC that performs the processing of the registered breathing signals.Smartphone/Tablet: The sensing system communicates wirelessly with a smartphone application that runs the processing algorithm ubiquitously.Cloud: Breathing signals are sent wirelessly to a remote server, which performs cloud computing.Embedded hardware: Processing is performed directly on embedded systems, which are located in or near the sensing unit package.

##### Energy Autonomy and Power Consumption

Regarding the power supply, systems can be categorized according to whether (1) they harvest part of the energy required for system operation, (2) they use rechargeable batteries, or (3) they are directly connected to a power source through a cable. This section analyses the first two categories in more detail since systems connected to a power source are of less interest as they have unlimited power availability.

**(1)** 
**Energy Harvesters**


Few were the studies found in the systematic searches conducted in this review that harvested energy [77,84,104]. However, some energy harvesting techniques have been reported experimentally in other wearable systems [240,241,242,243,244,245,246,247,248,249]. This section presents a description of these techniques and how they were implemented in the respiratory sensing systems. They were based on magnetic induction, piezo electric effect, triboelectric power generation, pyroelectric effect, thermoelectric effect, electrostatic power generation, and solar cells.
*Magnetic induction generator*: A small electric generator can be used to transform mechanical energy into electrical energy according to Faraday’s law. An electric current is induced in the generator coils by a changing magnetic field produced by the movement of the rotor due to the mechanical energy applied to it during breathing. The amount of generated voltage can be calculated according to Equation (1) [135].

(1)$$V=\;-\frac{N\times K_{1}}{K_{2}}\times \frac{d∆C_{Chest}}{dt},$$ where *N* is the number of turns of the coil, $∆C_{Chest}$ is the circumference change of the chest, *K*_2_ is the proportionality constant between $∆C_{Chest}$ and the angular displacement, and *K*_1_ is the proportionality constant between the magnetic flux and the rate of change of the angular displacement. The prototype presented by Padasdao et al. [135] attached the motor to a plastic housing with an armature fixed to the rotor gears (or shaft) (Figure 13A). A non-elastic wire was wrapped around the chest. One side of the wire was fixed to the plastic housing and the other end was attached to the armature. A piece of hard felt was fixed to the housing to help stabilize the device against the body. A spring was attached between the armature and the plastic housing to provide a restoring force to the armature. During inspiration, the non-elastic wire pulled the armature, leading to rotor rotation. During expiration, the spring pulled the armature back, leading to rotor rotation in the opposite direction. In this way, energy was harvested. In the work of Padasdao et al. [135], the electrical signal generated was used to obtain the RR instead of supplying power to the system. However, this is an example of how respiratory movements can be converted into electrical energy.

Other respiration-based energy harvesting systems can be found in the literature. The works of Delnavaz et al. [240] and Goreke et al. [241] used air flow to produce power with magnetic induction generators. On the one hand, the prototype of Delnavaz et al. [240] was made up of two fixed magnets located at the ends of a tube (opposite poles facing each other) with a free magnet inside the tube (Figure 13B). The free magnet was suspended due to the repulsive forces with the fixed magnets. A coil was wrapped around the outside of the tube. When a subject breathed into the tube, the free magnet moved around its static position. In this way, a voltage was induced in the coil since it was crossed by a variable magnetic field, which caused the magnetic induction. Experimental results showed that more than 3 µW were generated. The induced voltage in a closed circuit (*U*) was proportional to the magnetic flux gradient (dϕ/dx) and the velocity of the magnet (dx/dt), according to Equation (2).
(2)$$U=-N\frac{d\emptyset }{dt}\frac{dx}{dt}.$$

On the other hand, a microelectromechanical-scale turbine was presented by Goreke et al. [241]. The turbine had 12 blades on its outer contour and ball bearings around the center embedded in grooves (Figure 13C). A permanent magnet was integrated in the area between the ball bearings and the turbine blades. The entire prototype was encapsulated in a package with rectangular openings for the airflow. The prototype presented was under development and not fully implemented. The operating principle of the system could be as follows: by flowing air for the rectangular openings, the blades rotate and move the turbine in such a way that its coils see a variable magnetic field generated by the fixed magnet. This generates power through magnetic induction. The maximum power generated was 370 mW.*Piezoelectric energy harvesting*: These harvesters generate a voltage when compressed or stretched [242]. In the work of Shahhaidar et al. [242], they were embedded in a belt alongside the chest. Due to low capacitance of the piezoresistive materials, the overall harvested energy was low. Therefore, this piezoresistive configuration was unable to provide the necessary energy to power the entire system. The main drawback to adopting this energy harvesting technique for respiration sensors is that the required vibration frequency is much higher than the respiration frequency. In this sense, the paper of Li et al. [243] presented a prototype based on the interaction between a piezoelectric cantilever and a magnet placed on a substrate (Figure 14B). The vertical vibration of the cantilever due to the magnet presence allowed generating a constant amount of energy. The substrate with the magnet was attached to subject body (a limb joint). The movements of the subject led to substrate stretching and contraction, which caused the vibration of the piezoelectric cantilever. The energy generated was stable for different types of movements, since it was tested on different parts of the body. The energy harvester worked correctly for subject movements in the frequency range of 0.5–5.0 Hz. It has potential to be used with breathing movements. Meanwhile, Wang et al. [244] presented a piezoelectric rubber band that could be mounted on an elastic waistband to generate electricity from the circumferential stretch caused by breathing. The paper showed a structure made up of top and bottom electrodes with two solid layers and one void layer in between (Figure 14A). They were made of composite polymeric and metallic microstructures with embedded bipolar charges. Finally, the work of Sun et al. [245] presented an energy harvester from respiration air flow based on the piezoelectric effect. They used piezoelectric polyvinylidene fluoride (PVDF) microbelts that oscillated under low-speed airflow to generate electrical power in the order of magnitude of µW (Figure 14C).


*Triboelectric energy harvesting*: They generate charges by rubbing two different materials (one is an electron donor and the other is an electron acceptor), resulting in the creation of a potential in the contact region [250]. One possible setup is to attach the tribo-pair to a belt to detect variations in abdominal circumference. Triboelectric generators were used in breathing studies as a means of measuring RR, but not as energy harvesters, since the power generated is low for the power requirements of the entire respiration monitoring system that includes also a data transmission module. In the work of Zhang et al. [246] two belts (one extensible and one inextensible) were attached to each side of two materials (Figure 15A). A mechanical experiment was performed to obtain the peak voltage for different sliding amplitudes in the range of 2.5 to 30 mm that represents the typical displacement of a breathing depth. The result of this experiment was Equation (3).
(3)$$V_{peak}=0.01383X_{Max}+0.0092,$$ where Vpeak is the peak value of the voltage, and the $X_{max}$ is the maximum sliding displacement of the tribo-pair. A similar approach was proposed by Zhang et al. [77]. They presented a tribo-pair with both sides of one material fixed to two “Z-shaped” connectors that were attached to a belt with an inextensible part and an extensible part (Figure 15B). The abdominal contraction and expansion associated with respiration caused deformation of the two “Z-shaped” connectors. This deformation led to a process of contact and separation of the tribo-pair, generating an electrical signal. 


A self-powered respiratory sensor and energy harvester was also shown in the work of Vasandani et al. [247]. The working principle was very similar to the work of Zhang et al. [77] but, in this case, a prototype was built with movable and fixed supports (Figure 15C). The two materials were fixed to these two supports. The movements associated with respiration caused an angular displacement of the movable support by means of a belt and a lever mechanism, harvesting energy. The voltage obtained between the electrodes was zero in case of full contact and rose to 9.34 V for a 60° separation. The maximum area power density was 7.584 mW/m^2^.
*Electrostatic energy harvesting*: It is based on the change of parameters of a capacitive device, which is called electrostatic energy harvester. Breathing may cause separation of the capacitor plates or modification of the plate area, among others [251]. This energy harvesting technique is not common in respiratory systems. The prototype of Seo et al. [248] showed a capacitor made of two metal electrodes and an insulating layer in between. The capacitance of the prototype varied with respiration. This was because the area of the top electrode was variable depending on the presence of a wet surface associated with respiration (Figure 16). Humid exhaled breath air was cooled by the ambient air on the top surface of the insulated material. Thus, the water molecules were condensed, acting as part of the upper electrode and changing the capacitance of the prototype. This condensation provided a thick layer that became part of the electrode. Then, the water naturally evaporated due to its vapor pressure and the device returned to its original status. The variable capacitance allowed the charges to circulate, harvesting electrostatic energy. The prototype presented in Reference [248] reported a generated power of 2 μW/cm^2^.*Pyroelectric energy harvesting*: These harvesters are based on the reorientation of dipoles owing to temperature fluctuations [252]. Therefore, they need a temperature variation in time. Xue et al. [249] presented a prototype made of a pyroelectric component (metal coated PVDF film) covered with electrodes and mounted on the respirator of a mask at the location where air flows during breathing (Figure 17). The size of the prototype was 3.5 × 3.5 cm. The estimated current generated can be derived from the pyroelectric effect equation:

(4)$$I=Ap\frac{dT}{dt}$$ where *I* is the generated current, *A* in the sensing area, *p* is the pyroelectric coefficient (approximately 27 μC/m^2^ K), and *dT*/*dt* is the variation in temperature. Temperature variation is due to the difference between human body temperature and ambient temperature. It is also influenced by the transformation of water vapor into exhaled gas. The pyroelectric generator is heated by expiration and cooled by inspiration. Therefore, electricity is harvested from a change in temperature over time. Peak power reached up to 8.31 μW with an external load of 50 MΩ.


*Thermoelectric energy harvesting*: These harvesters are based on the Seebeck effect. They convert a temperature gradient into electric power. Therefore, they need a temperature variation in space [253]. A thermoelectric module is an array of p-type and n-type semiconductors. According to Nozariasbmarz et al. [252], the conversion efficiency of a thermoelectric generator can be calculated as:


(5)$$\eta =\frac{T_{H}-T_{C}}{T_{H}}\frac{\sqrt{\left(1+ZT\right)}-1}{\sqrt{\left(1+ZT\right)}+T_{C}/T_{H}},$$ where *T_C_* and *T_H_* are the temperature of the cold and hot sides, respectively. *ZT* is the dimensionless figure of merit for the thermoelectric module. For the thermoelectric material, *ZT* can be calculated according to:
(6)$$ZT=\frac{s^{2}\sigma }{k}T,$$ where *s* is the Seebeck coefficient, *σ* is the electrical conductivity, *k* is the thermal conductivity, and *T* is the absolute temperature.

Thermoelectric energy harvesters are not usually considered to power respiratory sensors. In the review of Nozariabsbmarz et al. [252], it was reported that several generators used the heat from the wrist for thermoelectric power generation.
*Solar cells*: This technology has been also used to power respiratory sensing systems. The energy produced by the solar cells is stored in a battery through a charge regulator that also controls the discharge of the battery to power the sensing system. The charge regulator controls that both the battery and the sensing system are supplied with adequate voltage and current levels. Figure 18 shows an example of sensing system powered by solar cells. Solar-powered systems have not been extensively explored in existing studies. As an exception, the work of Gorgutsa et al. [84] presented a Received Signal Strength Indicator through standard Bluetooth protocol using a hybrid-spiral antenna made of multi-material fibers. The system was integrated into a cotton shirt. They used a low-power Bluetooth module that was powered by a rechargeable battery and a solar cell on a custom printed circuit board.

**(2)** 
**Battery-Powered Systems**


Battery-powered systems require, at least, a battery and a charger. These two elements should be considered in the sizing of the system. Batteries are usually one of the most limiting components in terms of space (Figure 19).

Power autonomy determines the viability of a system. The autonomy of a battery-powered respiration sensing system is obtained by calculating or measuring its battery life, which is defined as the time that a system can operate with a fully charged battery. Two different factors must be determined when performing tests to measure battery life: system operating mode and the way of measuring battery life. 

Regarding system operating mode, there are essentially two different approaches: *Continuous operation*: Battery life is measured with the breathing device operating continuously.*Continuous operation + inactivity periods*: A typical daily use of the system is considered, which may include certain inactivity periods in which the device is in “idle” mode or even off (not used).

Regarding the way of measuring battery life, it should be noticed that it depends on the type of battery used and its parameters. The main parameter of a battery is its capacity, which determines the nominal amount of charge that can be stored. It is usually expressed in mAh. As a general rule, the higher the capacity, the longer the battery life. However, capacity depends on several external factors, such as discharge rate, operating temperature, aging, and state of charge (SOC). When a battery is discharged at low rate (low current), the energy is delivered more efficiently. Higher discharge rates (higher currents demanded by the breathing system) lead to a reduction in effective battery capacity [255]. Temperature also affects battery capacity in such a way that low temperatures decrease capacity. Aging may also decrease the capacity [256]. If a battery is not full, the state of charge (SOC) must also be considered. It represents the percentage of capacity that is currently available with respect to the rated capacity. 

The most common and sensible approach is that tests are conducted with a new fully-charged battery that operates in the nominal temperature range and discharges within the nominal current range. Under these conditions, the nominal capacity of the battery can be considered its true capacity. Otherwise, different reduction factors (<1) should be applied to rated capacity. Therefore, different ways to measure battery life experimentally can be found in existing studies:*Measure of battery life directly*: A battery can be considered discharged when the voltage drops below a certain value (3.6 V [257] for common small batteries). Therefore, by taking a full battery and monitoring the output voltage, it is possible to obtain battery life with expression (7).
(7)BatteryLife (h)=InitialTime−DischargeTime.

*Measure of current consumption*: Current consumption of the respiratory sensing system can be measured experimentally or estimated from the datasheets of the system components. The formula for calculating battery life is different for each operation mode:
○*Continuous operation*: The system is assumed to operate continuously consuming an average current value.
(8)Batery Life (h)=Capacity(mAh)·SOCfactor·Cfactor·Tªfactor·AgefactorOC (mA), where SOCfactor,  Cfactor,  Tªfactor, Agefactor∈ℝ[0,1] are reduction factors of the capacity to be applied in case tests are not performed under the optimal conditions mentioned above, and OC is the average value of the operating current.○*Continuous operation + inactivity periods (rough estimate)*: Current consumption in the operation and inactivity periods is assumed to be “constant”.
(9)Batery Life (h)=Capacity(mAh)·SOCfactor·Cfactor·Tªfactor·AgefactorOC (mAh)·nminOCnmintotal+IC (mAh)nminICnmintotal, where *IC(mAh)* is the average value of current consumed by the system in idle or non-active modes, nminoc is the number of minutes that the breathing system is in operation mode during a certain period of time (for instance, one day), nminIc is the number of minutes that the breathing system is in idle or non-active modes for the same time period, and nmintotal=nminOC+nminIC.○*Continuous operation + inactivity periods (fine estimate)*: The calculation of battery life is performed using a more accurate model. Different values of current consumption are considered in operation and inactivity modes. In this calculation, the system can adopt not only two states, but *n* states. Let c=[c1,c2,…,cn] be the average current values of each of the *n* different states of the respiratory system considered, and nmin=[nmin1,nmin2,…,nminn] the number of minutes in a given period of time (for instance, one day) that the breathing system remains in each state of the *n* possible states. The calculation can be done with Equation (10).
(10)Batery Life (h)=Capacity(mAh)·SOCfactor·Cfactor·Tªfactor·Agefactor∑i=1nci·nmini∑j=1nnminj.

#### 3.2.2. Results of the Analysis

The previously described items were analyzed for the studies found as a result of the systematic review. These items were the use of wired or wireless data transmission, the performance of centralized or remote processing, the specific station used to carry out processing and the energy autonomy of the prototypes. They were studied for the wearable category as these elements are limiting in non-contact sensing systems. However, they are less crucial in environmental systems, since most of them use wired communications and are connected to a power source.

Table 4 shows a comparison of the approaches found in the state of the art for the wearable group. The first two columns of Table 4 show the specific studies that used wired and wireless data transmission, and Figure 20 presents the percentage distribution of the type of transmission. The use of wired and wireless technologies was similar. 

Figure 21 shows the distribution of wireless technologies used for data transmission. Bluetooth was the preferred technology, as it is suitable for applications that send point-to-point information over relatively short distances and require high-speed data transmission. Its main drawback is power consumption, which could be a limitation for continuous monitoring, as existing studies state that the battery life is not more than a few hours. However, in view of Table 4, this method seems suitable for many applications. Wi-Fi, radio frequency, or Zigbee were used in a limited number of studies [73,96,144,156,159]. Regarding wired transmission, third column of Table 4 shows that USB communication was the preferred option [74,76,86,87,88,109,114,118,133,141,158].

Once measurements are transmitted, a main station processes them. Figure 22 shows the percentage distribution of the processing stations used in the studies selected in the systematic searches. PCs were the preferred processing stations, showing that most authors performed centralized processing, while the use of smartphones, tablets or cloud computing was not so common [2,59,65,66,67,74,76,77,84,91,98,99,101,102,107,109,116,119,122,130,132,134,143,144,156], although they were found in 30% of studies.

Regarding energy autonomy of systems, the use of energy harvesters was residual [84,104], which can be due to the fact that studies presented complete systems that included data transmission and processing modules. These modules are energy demanding, and therefore the use of energy harvesters can only be used as a complement, but not as the primary power source. In this regard, many studies [2,3,17,62,65,73,84,86,87,89,91,98,99,101,114,115,116,119,131,144,145,147,162] used rechargeable batteries to power the systems. The most common declared battery lives were in the order of hours (Figure 23) [2,17,62,69,101,115,119], although some studies did not even provide data on this point.

There were a set of studies focused on minimizing power consumption. They included low power data transmission technologies. In this regard, Milici et al. used wireless transponders [91] to obtain autonomy of more than one year, while Mahbub et al. [98] adopted Impulse Radio Ultra-Wideband (IR UWB), which led to an autonomy of about 40 days. In general, battery live is highly dependent on transmission technology. The works of Bhattacharya et al. [156], Puranik et al. [73], White et al. [96], Ciobotariu et al. [144], and Mitchell et al. [159] used wearable devices with Wi-Fi [73,96,144,156], Zigbee [159], or GSM/GPRS [144], with high variability in power consumption.

### 3.3. Validation Experiments

#### 3.3.1. Items of Analysis

Different items were considered to analyze the validation experiments carried out in the studies:Subjects: Almost all studies used volunteers to assess the respiration sensing systems. In this case, it is required to provide data, such as the number of subjects who participated in the tests and their main characteristics (age, weight, height, sex, and health status). As breathing studies generally involve humans, it is mandatory to have the approval of the competent ethical committee (following the Declaration of Helsinki [258]) to recruit the subjects to participate in the study, to inform them about the study, and to obtain their consent.Activities/positions: This item refers to the specific activities or positions that volunteers who participate in the tests are asked to perform as part of the validation experiments. The most common positions adopted in existing studies are represented in Figure 24 with an example sensor.Whether or not motion artifacts are included in the different activities.Number and values of RRs or volume rates to be tested in the experiments.Number of repetitions of the different test scenarios.Duration: The designed tests (activities and positions, number of RRs or volume rates, and number of repetitions) determine the duration of the experiments.Experiment design: This item refers to the strategies adopted to validate the breathing sensors. Three main methods have been found in the state of the art (Figure 25):

**Artificial validation prototypes:** Some studies used artificial prototypes that emulated human conditions rather than real volunteers. On the one hand, if the sensor were worn on the chest, diaphragm, or thorax, a mechanical structure that emulated human respiration could serve for validation. That was the approach adopted by Padasdao et al. [135]: a motor moved a mechanical chest to the rhythm and depth of human breathing (Figure 25A). Similarly, the work of Witt et al. [141] also used a mechanical chest driven by a stepper motor, setting the amplitude and frequency of the movements to simulate breathing activity. Another set of works [77,94,110,114,146] used machines or custom prototypes that applied traction and compression movements to simulate human respiration on strain sensors. On the other hand, if the system is to be worn in the nose or mouth, an artificial prototype can be built that emulates the airflow associated with respiration. For that, Agcayazi et al. [123] used a mannequin equipped with an inflatable cuff bladder that emulated breathing cycles, which is similar to the prototype of Koch et al. [90]. For humidity sensors, authors designed controlled humidity chambers using humidifiers and dry air compressors [74] or switches for controlling nitrogen flow and a motor to control the dispersion of water vapor [97]. Finally, other studies presented artificial validation prototypes adapted to the specific sensors used for respiration monitoring. Zito et al. [226] validated a radar sensor with a moving target that emulated the movements associated with breathing.

Validation using artificial prototypes has the advantage that different respiration or volume rates can be programmed precisely. These theoretical values can be compared with the measurements obtained with the sensor. Thus, no error can be attributed to the validation method. A typical validation workflow using this method is outlined in Figure 26. In this method, sensor measurements may be contained in matrix  A=(aij)ϵℝk×m, where *k* is the number of repetitions per parameter, and *m* is the number of different parameter values to evaluate. This measurement matrix **A** can be compared with the reference matrix B=(bij)ϵℝk×m. Matrix **B** contains the reference values used to program the artificial validation prototype. Therefore, all the elements in a given row have the same value as the *j*th reference parameter (column) remains the same for all repetitions (∀iϵ[1..k], row).

**Metronome as reference:** When humans are involved in the validation experiments, one option is to use a metronome to set the rate of respiration that subjects must follow during the tests (Figure 25B). The advantage of this method over artificial prototypes is that the sensing system is tested with the target subjects and not with an emulation of a human chest or throat. However, its weak point is that subjects may not accurately follow the rate of the metronome. Therefore, part of the measurement error can be attributed to the test design itself rather than to the sensing system. A typical validation workflow using a metronome as a reference is summarized in Figure 27. The measurements recorded by the sensor under validation may be contained in matrix A=(adefgh)ϵℝn×k×p×l×m, which is a five-dimensional matrix with the measured values for each subject *d*, repetition *e*, activity *f*, position *g* and parameter value *h*. The reference to compare **A** is matrix B=(bdefgh)ϵℝn×k×p×l×m, which contains the reference breathing parameters set in the metronome for each subject *d*, repetition *e*, activity *f*, position *g* and parameter value *h*. Therefore, **B** exclusively contains the values of vector z=[z1,z2,…zm], which are the possible settings for the metronome (Figure 27).

**Validation against a reference device:** The most complete way to validate a new sensor is to compare its performance with the performance of a reference sensor considered as a gold standard (Figure 25C). The reference sensor and the sensor under validation must be worn at the same time to obtain synchronized measurements. Having synchronized measurements allows the sensing capabilities of both sensors to be compared fairly. The sensor under test should provide measurements as close as possible to those of the reference sensor. It is important to note that the reference sensor also has a measurement error. Therefore, this error should be considered in the comparison, as it may influence the results. Respiratory values provided by the reference device may differ slightly from real values. This validation method faces several challenges. First, it is essential to synchronize both measuring instrument and this synchronization can be difficult. Second, most commercial products do not provide information on how the final breathing parameter (RR or volume parameter) is obtained, so the comparison of measurements may not be obvious. In addition, most products do not allow selecting the refresh time window or do not even provide information about the length of this window, so it is not possible to know the set of measurements used to calculate the output respiration values. Figure 27 shows a typical block diagram of the validation method when using a reference device. The results of this validation are matrices A=(adefgh)ϵℝn×k×p×l×m and C=(adefgh)ϵℝn×k×p×l×m, which contain the measured values for each subject *d*, repetition *e*, activity *f*, position *g,* and parameter *h* for the sensor under evaluation and for the reference sensor, respectively.

#### 3.3.2. Results of the Analysis

Table 5 presents the results of the analysis of different items of the validation experiments for both wearable and environmental systems. Large differences among studies were observed in all aspects of the experiments: protocol, number of subjects, positions, types of breathing, duration, and inclusion of motion artifacts. 

In relation to the number of subjects involved in the tests, 71% of the studies that provided this data included 10 subjects or less. Only 13% of the studies included more than 20 subjects [7,10,19,64,81,132,135,147,173,175,211,232]. There were also a considerable number of studies (53) that did not even provide this information. A part of them did not use subjects for sensor validation. 

Regarding the duration of the experiments, most of the studies carried out short experiments of a few minutes. In fact, 58% of the studies performed tests of less than 5 minutes [69,70,81,86,87,88,96,100,102,104,111,125,136,161,163,165,171,172,173,174,193,195,196,200,212,215,221,228,229,232]. Most of the works that conducted longer tests included sleep studies [7,17,53,60,115,146,148,164,165,169,173,192,198,205,211,220,223]. Twenty-six studies reported that motion artifacts were considered during testing. They showed that the inclusion of motion artifacts in experiments greatly influenced sensor performance [2,9,17,53,61,62,66,67,68,81,108,109,117,119,131,132,135,147,157,178,187,190,196,205,210,221,225]. In relation to the activities or positions considered in the experiments, lying down and sitting were the most tested positions. Other positions or activities, like standing, walking, moving, or running, were used in a minority of studies [2,17,21,61,62,66,67,68,77,79,81,91,94,101,102,103,108,110,111,115,118,119,124,129,131,132,135,146,147,148,149,177,178,188,205,214,233,235]. Most of the studies that provided information on activities considered more than one position [2,6,9,17,21,52,53,61,62,66,67,68,77,79,85,86,87,88,94,102,108,110,115,118,119,124,129,131,132,133,135,146,147,148,157,164,165,169,171,177,178,187,196,198,205,210,211,213,214,220,221,223,225,233,235]. It was also common to test different values of the respiration parameter (for example, RRs from 10 to 22.5 bpm in the study of Vanegas et al. [254]). 

In relation to the validation protocol, Figure 28 shows the distribution of the analyzed studies in the three categories introduced in Section 3.3.2: validation with an artificial prototype, metronome as reference and validation against a reference device. A new category was created to cover studies that performed informal validation. It was called “human observation”, since an expert provided a value of the breathing parameter from direct observation of the signals recorded by the sensors. Figure 28 shows that validation using a reference device was the predominant approach (adopted by 67% of the studies that performed validation), followed by the use of an artificial validation prototype (10%) [69,74,77,90,93,97,110,114,116,119,123,135,141,146,150,226]. It is also worth noting that 53 studies presented the sensing systems without providing evidence of their validation.

### 3.4. Sensor Measurement Processing

#### 3.4.1. Items of Analysis

This category includes the following items: performance evaluation, software used for the analysis, and processing algorithm. This section describes them in detail.

##### Performance Evaluation

The evaluation of sensor performance can be done using several figures of merit, such as absolute error, relative/percentage error, root mean square error, correlation factor, Bland-Altman plot, calculation of accuracies, or linear regression. 

**Absolute error (Δ):** Difference between the value measured by the sensor under test (*x*) and the reference value (*y*). It is calculated according to Equation (11).
(11)Δ=x−y.

It is more common to provide the mean absolute error (MAE) as the mean of the absolute value of all absolute errors: (12)$$\mathrm{MAE}=\frac{1}{n}\sum_{i=1}^{n}\left|x_{i}-y_{i}\right|,$$
where *n* is the number of measurements obtained from the sensor under test, *x_i_* the values of those measurements, and *y_i_* the reference values associated with those measurements for the “artificial validation prototype” method and the “metronome as reference” method or the measurements of the reference device for the “validation against a reference device” method. 

**Relative error (RE):** Absolute error of the breathing sensor under test (Δ) divided by its reference (true) value (*y*). Thus, it provides an error value relative to the size of the breathing parameter being measured. It can be obtained according to Equation (13). The mean of the relative errors (MRE) can be obtained using Equation (14).
(13)$$\mathrm{RE}=\frac{\Delta }{y},$$
(14)$$\mathrm{MRE}=\frac{1}{n}\sum_{i=1}^{n}\frac{\left|x_{i}-y_{i}\right|}{yi},$$
where *n*, *x_i_*, and *y_i_* are the same parameters as for the MAE.

If the relative error is expressed as a percentage, it is called the percentage error, although many authors also provide the relative error in percentage.

**Root mean square error (RMSE):** In respiration sensing studies, it is also used to compare the difference between the values measured by the sensor under analysis and the reference results. It is the root mean of these differences and can be obtained according to Equation (15).
(15)$$\mathrm{RMSE}=\sqrt{\frac{\sum_{i=1}^{n}{\left(x_{i}-y_{i}\right)}^{2}}{n}},$$
where *n*, *x_i_*, and *y_i_* are the same parameters as for the MAE. 

**Correlation factor:** It provides a measure of the relationships between the measurements taken by the respiration sensor under test and the reference data. There are different ways to calculate this correlation factor. Pearson correlation factor is one of most extended (Equation (16)).
(16)$$\gamma_{xy}=\frac{n\sum x_{i}y_{i}-\sum x_{i}\sum y_{i}}{\sqrt{n\sum x_{i}^{2}-{\left(\sum x_{i}\right)}^{2}}\sqrt{n\sum y_{i}^{2}-{\left(\sum y_{i}\right)}^{2}}},$$
where *n*, *x_i_*, and *y_i_* are the same parameters as for the MAE. A correlation factor of 1 means maximum agreement between measured and reference data (optimal case), while a factor of 0 means that there is no relationship between the datasets.

**Bland-Altman analysis**: It is a graphical method to compare the measurements from the breathing sensor under test with the reference breathing values. A scatter diagram is drawn with the horizontal axis representing the mean between measured values and reference values ($\frac{\left(x_{i}\;+\;y_{i}\right)}{2}$) and the vertical axis representing the difference between those values (*x_i_* − *y_i_*). In addition, a horizontal line is included in the plot with the mean value of all differences. Two more horizontal lines (one upper and one lower) are plotted representing the limits of agreement (±1.96 times the standard deviation of the differences). The Bland-Altman plot is useful to show relationships between the magnitude of the breathing parameter and the differences between measured values. It may also help to identify systematic errors in measurements or to detect outliers, among others. This method is especially suitable for the validation method in which the sensor under evaluation is compared to a reference device.

**Accuracy**: It is the proportion of true results with respect to the total number of samples [259]. It can be used in studies of respiration sensors that identify breathing patterns within a given set of *k* possible patterns. It can also be applied to studies that determine the value of a breathing parameter within a discrete set of *k* possible values. Let $x=\left[x_{1},x_{2},\ldots ,x_{n}\right]$ be the values of the *n* measurements taken by a respiration sensing system or the *n* labels of the breathing patterns recognized by the system. Suppose that, from the *n* different samples, *m* samples are correctly identified or measured, since they belong to the correct class of the *k* possible classes. Therefore, *(n-m)* samples are not classified correctly. The accuracy of the breathing system can be obtained as:(17)$$Accuracy\;\left(\%\right)=\frac{m}{n}\cdot 100.$$

**Linear regression:** It models the relationship between the values measured by the respiration sensing system under test (dependent variable) and the reference measurements (independent variable) by fitting a linear equation. The equation to fit has the form of y=a+bx, where *y* is the dependent variable, *x* is the independent variable, *b* is the slope of the line, and *a* is the intercept (value of *y_i_* when $x_{i}=0$). This linear fitting is performed using $x=\left[x_{1},x_{2},\ldots ,x_{n}\right]$, which is the set of *n* reference values of the breathing parameter, and $y=\left[y_{1},y_{2},\ldots ,y_{n}\right]$, which is the set of *n* values of the parameter measured by the sensing system under evaluation. In these conditions, the values of each *x_i_* and *y_i_* should be as close as possible ∀i∈[1..n]. This means that, if the match between the reference values and the measured values was perfect, the linear model should be a line with and intercept of 0 and a slope of 1. 

In addition, the coefficient of determination $r^{2}$ could also be calculated to obtain what percentage of the variation in the values measured by the sensing system are predictable from the variation of the reference values according to Equation (18).
(18)$$r^{2}=1-\frac{SE_{res}}{SE_{\bar{y}}},$$
where $SE_{\bar{y}}=\sum_{i=1}^{n}{\left(y_{i}-\bar{y}\right)}^{2}$ is the sum of the squares of the difference of each measured value *y_i_* with respect to the mean value of all measurements $\bar{y}$, and $SE_{res}=\sum_{i=1}^{n}\left(y_{i}-\left(a+bx_{i}\right)\right)\;$is the sum of the squares of the difference of each measured value *y_i_* with respect to the value predicted by the model. If $SE_{res}\;$is small, it means that the line is a good fit, and $r^{2}$ will be close to 1. Otherwise, if $SE_{res}\;$is large, it means that the difference between the measured values *y_i_* and the line is large, and $r^{2}$ will be close to 0 (bad linear fit). If the breathing system measured exactly the same values as the reference system, $SE_{res}$ would be zero and $r^{2}$ would be 1, which would be the ideal case.

##### Analysis Software

The most common tools used to analyze the measurements recorded by the sensors are: MATLAB: Popular numerical computing environment and programming language that is suitable for the implementation of algorithms, matrix operations, or data plotting, among others.Labview: System engineering software for applications that require testing, measurements, control, fast hardware access, and data information.Others: An extensive set of tools has been used in existing studies, such as Python (high-level, programming language specially focused on facilitating code readability), R (free software environment and programming language for statistical computing [260]), C# (general-purpose programming language developed by Microsoft [261]), C (general-purpose programming language that supports structured programming), OpenCV (open source software library for computer vision and machine learning [262]), Blynk (Internet of Things platform), Kubios HRV (heart rate variability analysis software for professionals and scientists), Audacity (free open-source audio software), Kinect SDK (suitable for developing gesture or voice recognition applications, using Kinect sensor technology [263]), LabChart (physiological data analysis and acquisition software [264]), Acqknowledge (software to measure, transform, replay and analyze data [265]), mobile/Android (mobile operating system), LabWindows/CVI (software development environment specially focused on measurement applications [266]), microcontroller/microprocessor (suitable if the processing is not done in any external software, but directly in the same microprocessor or microcontroller that controls the sensor), or custom applications (PC applications in which the native source could not be accurately determined).

##### Processing Algorithm

A broad set of algorithms has been used to process measurements from respiration sensors such as peak detection, maximum-minimum detection, detection of zero-crossings, threshold detection, frequency analysis, wavelet transform, or Kalman filter, among others. They are briefly described in this subsection.

**Peak detection:** It is based on the detection of peaks in the signals registered by the sensing system (Figure 29). If no restriction is imposed regarding peak prominence, a peak can be calculated on a signal x=[x1,x2,…,xn] according to Equation (19), where *n* is the number of samples in the signal. However, this method is extremely sensitive to noise and fluctuations (Figure 29A). To improve detection, it is possible to set a minimum surrounding number of samples (*p*) in which the values must be below the peak value (Equation (20)) to accept the detected peak (Figure 29B). Another option is to impose a strictly increasing slope on the *p* samples preceding the peak and/or a strictly descending slope on the *p* samples after the peak (Figure 29C), according to Equation (21).
(19)xi−1<xi>xi+1   ∀i∈ℤ:i∈[1,n],
(20)xj<xi>xh    ∀j∈ℤ:j∈[i−p,i−1]  ;  ∀h∈ℤ:h∈[i+1,i+p],
(21)xj−1<xj    ∀j∈ℤ:j∈[i−p,i]  and   xh+1<xh  ∀h∈ℤ:h∈[i,i+p].

The peak detection method to process respiration signals has several important parameters that determine the number of detected peaks. Peaks selected according to Equations (19), (20), or (21) can be classified according to the prominence of the peak, discarding those peaks that are below a threshold value to avoid the effect of noise and fluctuations. Peak prominence can be defined as the vertical distance between the closest local minima (in horizontal direction) and the peak, although there are other possible definitions [267]. Let $y=\left[y_{1},y_{2},\ldots ,y_{n}\right]$ be a vector containing the magnitude of all local minima of signal *x*, and $b=\left[b_{1},b_{2},\ldots ,b_{n}\right]$ the position (horizontal value) of the local minima *y*. If *a_i_* is the position of peak *i*, and *b_k_* is an element of *b* that satisfies that $b_{k}=\min\left|b-a_{i}\right|$ (the position of the local minima closest to *i*), then, a peak *i* will only be accepted if its prominence is above a set threshold (*PP*), $\left|x_{i}-y_{k}\right|<PP$ (Figure 29D).

Another parameter that may be used to determine the number of peaks is the distance among them. Breathing signals are low frequency (usually less than 25 bpm [254]); therefore, a threshold (TD) is generally established to discard those peaks that do not differ by, at least, TD from another previously detected peak (Figure 29E). Let c=[c1,c2,…,cq] be the position of the *q* peaks detected in a signal. A new peak candidate *i*, with position on the horizontal axis *d_i_*, will only be accepted if |di−cj|<TD , ∀j∈ℤ:j∈[1,q].

It is also common to discard peaks that do not reach a certain level TL (xi<TL) (Figure 29F) or, alternatively, that a new peak *i* is discarded if its value does not differ a given threshold TV from the *q* peaks already detected; that is, if |xi−xj|<TD , ∀j∈ℤ:j∈[1,q] (Figure 29G).

**Maximum-minimum detection:** A popular processing technique is to identify maximum and/or minimum points in the breathing signals (*x*). Massaroni et al. [103] used the maximum and minimum values to obtain the respiratory period (*T_r_*), as well as inspiratory (*T_i_*) and expiratory (*T_e_*) time. The process for detecting maximum and minimum points is similar to peak detection. 

**Zero-crossings:** Technique based on the detection of the crossings of a breathing signal by a “zero” level taken as a reference. Given a respiration signal composed of *n* values$\;x=\left[x_{1},x_{2},\ldots ,x_{n}\right]$, a new zero crossing at the *i* value is detected when inequality (22) is satisfied.
(22)$$x_{i-1}<x_{i}<x_{i+1}\;\;\;\;i\in \left[1..n\right].$$

One of the challenges of this method is to find the “zero” level taken as a reference to detect the crossing. One possible option is to detect the maximum and minimum values in a specific window and obtain the “zero” level as the mean of those values (max(x)+min(x))/2. However, this method is sensitive to outliers (Figure 30A). A possible solution is to take the median of *x* as the “zero” level (Figure 30A). Another option is to remove 10–20% of the largest and smallest values of *x*, obtaining a subset of values *y* ⊆ *x*. Then, the “zero” level can be calculated as the mean of the maximum and minimum values of *y*.

The “zero-crossings” technique is also affected by trends or biases in the measurements. Trends may be due to movement of the sensing element or movement of the subject. It is a common phenomenon, especially in belt-attached breathing sensors. Figure 30B shows a real breathing signal with trends (blue curve) from a public dataset [254]. If the “zero” level is calculated on a signal with trends, many crossings may go undetected since the same “zero” level is not a valid reference for the entire signal. To solve this problem, it is possible to eliminate trends in the signal by subtracting the bias (Figure 30B, orange signal). Another option could be to split the signal into shorter windows and calculate a different “zero” level for each window (Figure 30C). 

This technique is also sensitive to noise since the number of zero-crossings may increase in noisy signals. Figure 30D shows how noise is confused with multiple crossings at the “zero” level in a breathing signal. This can be avoided by defining a minimum distance in the horizontal direction (TD). Let z=[z1,z2,…,zq] be the positions in the horizontal axis of *q* detected “zero-crossings”; then, a new “zero-crossing” *i* with position *d_i_* will only be considered if |di−zj|<TD , ∀j∈ℤ:j∈[1,q].

**Threshold detection:** This technique is similar to “peak detection”, “maximum-minimum detection” or “zero-crossing detection”. In this case, the level to detect is not a characteristic point of the curve but a certain threshold value. The same analysis performed for the previous categories could be applied to this method.

**Frequency analysis:** This category includes different techniques that make use of frequency information to obtain respiration parameters. The most common approach is to use the well-known Fourier Transform (FT). Several studies detected peaks in the spectrum of respiration signals or in their power spectral density (PSD) to obtain the breathing parameters. This method depends on the time window (Figure 31A). To provide meaningful data, long time windows are desirable. However, this limits refresh time of the system. A compromise between accuracy and refresh time is required. Figure 31A shows a breathing signal and its spectra obtained with the FT for different refresh time windows. The example respiration signal has a frequency of 0.33 Hz (20 bpm) and a sampling frequency of 50 Hz. For a 4-s time window, the maximum available resolution is $\;F_{s}/N$, that is, 0.25 Hz. Figure 31A shows that the detected frequency is in the range of 0.25–0.5 Hz. This resolution is 0.125 Hz for the 8-s time window (frequency detected in the 0.25–0.375 Hz range) and 0.0625 Hz for the 16-s time window (frequency detected in the 0.3125–0.344 Hz range). It can be seen that the wider the time window, the more accurate results are obtained using this method. However, wide time windows make it difficult to apply respiration monitoring systems to critical scenarios where instantaneous values must be provided.

This transform is also sensitive to noise fluctuations. Noise fluctuations are generally of a much higher frequency than breathing signals. Therefore, it is common to pre-filter the signals to remove frequencies that exceed those of breathing activities. Figure 31B shows the frequency spectrum of a real respiration signal without filtering and with digital low-pass filtering. It can be seen that the peak of the respiration frequency (0.33 Hz) is more separated from the rest of the spectrum values in the filtered signal (7 units for the filtered signal and 5 for the unfiltered signal). If noise levels increased, it would even be difficult to distinguish the peak associated with the respiration frequency.

On the other hand, low frequency signal fluctuations may appear due to movements in the sensing device or movements of subjects if breathing is measured during dynamic activities, such as walking. These fluctuations must be treated to provide accurate results. They can be mathematically modeled according to Equation (23).
(23)$$v\left(t\right)=A\left[1+\lambda \sin\left(w_{f}t\right)\right]\sin\left(wt-\varphi \right),$$
where *w* is the angular frequency of the normal breathing signal, *w_f_* represents the angular frequency of the interference-causing activity, and *λ* is the magnitude of that activity. Figure 31C shows an example of a real breathing signal with low frequency fluctuations. Those frequency fluctuations can lead to peaks at very low frequency values of the spectrum. As low-pass filters are generally applied, those frequencies would not be removed and could therefore be confused with the respiration parameter, which is also low frequency. 

Sudden movements of subjects may also cause fluctuations in signals, which can affect the measurements. Figure 31D shows an example of a real respiration signal with fluctuations due to movements during acquisition tests. The bottom of Figure 31D shows its spectrum with a peak in the respiration frequency and other lower peaks (in red) at close frequencies due to signal fluctuations.

Other studies have also obtained breathing parameters from frequency using frequency modulation (FM) or amplitude modulation (AM).

**Wavelet transform:** It is used to decompose the breathing signal in such a way that a new representation can be obtained that allows a better detection of respiration peaks or crossings. It has been used in the continuous or in the discrete form [268]. In the continuous wavelet transform (CWT) a comparison is made between the respiration signal and an analyzing wavelet *ψ*. The wavelet is shifted by applying a dilation factor (*b*) and is compressed or stretched by applying a scale factor (*a*). Therefore, the CWT can be calculated according to Equation (24).
(24)$$CWT\left(a,b\right)=\int_{-\infty }^{\infty }x\left(t\right)\psi_{ab}^{*}\left(t\right)dt,$$
where *x*(*t*) is the breathing signal under analysis, and *∗* denotes the complex conjugate [269]. The scale factor (*a*) has an inverse relationship with the frequency (the higher the value of *a*, the lower the frequency, and vice versa). The dilation factor (*b*) allows delaying (or advancing) the wavelet onset. Therefore, it contains time information. In this way, the CWT can provide a kind of time-frequency representation where high frequency resolution is obtained for low frequencies and high time resolution is obtained for high frequencies. This is shown in Figure 32A where a real respiration signal is processed with the CWT. The time-frequency representation of the processed signal is shown in Figure 32A (right). It can be seen that a low frequency value around 0.33 Hz is identified with high resolution in frequency but low resolution in time. In the example respiration signal, the frequency remains fairly constant around the value of 0.33 Hz.

A variant of the WT is the multiresolution analysis (MRA) [269]. The MRA represents the voltage signal at different resolution levels by progressively analyzing the breathing signals into finer octave bands (Figure 32B). For that, the original signal is convolved with high and low pass filters that represent the prototype wavelet. The outputs of the low pass filter are called “approximation coefficients”, while the outputs of the high pass filter are called “detail coefficients”. Approximation coefficients are down-sampled by a factor of 2 and are again subjected to high-pass and low-pass filtering, obtaining a new set of “detail” and “approximation” coefficients. This process is repeated iteratively, resulting in different resolution levels. For a given decomposition level *n*, the detail coefficients contain information on a particular set of frequencies (from $f_{s}/2^{n}\;$to $f_{s}/2^{n+1}$), with *f_s_* being the sampling frequency. Regarding the “approximation coefficients” of the same decomposition level, they contain low-frequency information in the range $f_{s}/2^{n+1}-0\;$Hz. The number of decomposition levels of the MRA depends on the specific breathing signal, so the band of the respiration frequencies can be correctly identified. It is affected by the sampling frequency of the system. This decomposition process is explained graphically in Figure 32B. The original respiration signal (*x*) can be reconstructed from its detail and approximation coefficients as follows:(25)$$x=\sum_{j=1}^{l}D_{j}+A_{l},$$
where *l* is the number of decomposition levels. Figure 32B also shows an example of this technique applied to a breathing signal with a sampling frequency of 50 Hz. Six decomposition levels were selected to obtain five sets of detail coefficients in the ranges 25–12.5 Hz, 12.5–6.25 Hz, 6.25–3.125 Hz, 3.125–1.563 Hz, 1.563–0.781 Hz and one set of approximation coefficients in the range 0.781–0 Hz. The first and third levels of detail coefficients and the sixth level of approximation coefficients were represented in Figure 32B as an example. In this case, the level of interest was the sixth (approximation coefficients) since breathing signals are of low frequency. The Fourier Transform was performed on the coefficients of the sixth level, obtaining a clear peak at the frequency of 0.33 Hz, which matches the breathing frequency of the sample respiration signal (20 bpm). 

In the work of Scalise et al. [232], the signal was decomposed into 12 levels and level 11 was considered to obtain the RR. Guo et al. [166] performed a 4-level decomposition, selecting level 3 to calculate the RR. Therefore, the wavelet transform is used to obtain the respiration signals in the desired frequency band.

**Kalman filter:** This technique has been used by several studies as a sensor fusion method. Thus, it is not a method to extract breathing parameters but to fuse measurements from different sensors. When multiple respiration sensors are available, the measurements they provide are not exactly the same. Furthermore, measurements always contain noise. The Kalman filter is used to provide a final value based on the measurements of the different sensors, the model of variation of the breathing parameter, the noise model of the sensors, and the variation model [270]. Figure 33 shows an overview of the Kalman filter algorithm adapted to the fusion of breathing sensors.

The Kalman filter has two distinct phases: prediction and update. The prediction phase estimates the state (breathing parameter) in the current time step using the state estimate from the previous time step (previous breathing parameter). The breathing parameter predicted in this phase is called the “a priori” state estimate ${\hat{x}}_{k}^{-}$ and is obtained according to Equation (26).
(26)$${\hat{x}}_{k}^{-}=A{\hat{x}}_{k-1},$$
where ${\hat{x}}_{k-1}$ is the state estimate in the previous state, in this case the previous breathing parameter estimated, and *A* is the state transition model. Matrix *A* represents the expected evolution of ${\hat{x}}_{k-1}$ for the next transition. As breathing does not vary much in the short term [102], a common approach is to define *A* as an identity matrix, so the “a priori” state estimate ${\hat{x}}_{k}^{-}$ is equal to the previous state ${\hat{x}}_{k-1}$. If respiration is not expected to be constant in the short term, *A* should contain the linear variation model. The “a priori” estimate covariance $P_{k}^{-}$ (Equation (27)), which is a measure of the accuracy of the “a priori” state estimate ${\hat{x}}_{k}^{-}$, must also be predicted. It depends on the transition model *A*, the value of the covariance in the previous transition $P_{k-1}$, and *Q*. *Q* is the covariance of the process noise (the noise of ${\hat{x}}_{k}^{-}$ prediction model). In order to apply the Kalman filter, the process noise must follow a Gaussian distribution with zero mean and covariance Q_k_(~N(0,Q)). Although *A* and Q can vary at each time step *k*, it is common for them to take a constant value. Many methods exist to determine Q. In the breathing system presented by Yoon et al. [131] Q was a diagonal matrix (which is a common approach) of the order of 10^−4^.
(27)$$P_{k}^{-}=AP_{k-1}A^{T}+Q.$$

Once the “a priori” state estimate ${\hat{x}}_{k}^{-}$ has been predicted, the update phase comes into play. In the update phase, the “a priori” state estimate ${\hat{x}}_{k}^{-}$ is refined using the measurements $y_{k}$ recorded by the sensors. The result is the final value of the breathing parameter ${\hat{x}}_{k}$, which is called the “a posteriori” state estimate (Equation (28)).
(28)$${\hat{x}}_{k}={\hat{x}}_{k}^{-}+K_{k}\left(y_{k}-H{\hat{x}}_{k}^{-}\right).$$

The estimation of ${\hat{x}}_{k}\;$depends on the predicted “a priori” state estimate ${\hat{x}}_{k}^{-}$, the measurements registered by the different breathing sensors $y_{k}$ and the matrices K_k_ and H. K_k_ is known in the Kalman filter as the optimal Kalman gain. It minimizes the “a posterior” error covariance. A common way to calculate it is according to Equation (29).
(29)$$K_{k}=\frac{P_{k}^{-}H^{T}}{HP_{k}^{-}H^{T}+R}.$$

This gain depends on the “a priori” estimate covariance $P_{k}^{-}\;$and two model parameters (*H* and *R*). *H* is the observation model that relates the measurements taken by the sensors $y_{k}$ to the state space $x_{k}$ (breathing parameter), as follows $y_{k}=H{\hat{x}}_{k}$. It is common that previous techniques introduced in this section (peak detection, maximum-minimum detection, zero-crossings, threshold detection, frequency analysis, or wavelet transform) are used to directly estimate the respiration parameter from the measurements. In that case, the measurement space and the state space are the same. Thus, *H* could simply be the identity matrix. If the respiration parameter (RR, for example) were not provided directly as a result of the measurements, and other parameters were given instead (such as the number of peaks, zero-crossings, etc.), matrix *H* would contain the equations to calculate the breathing parameter from those values. Those equations were previously introduced in this section. 

*R* is the covariance of the observation noise (the noise associated with the measurements $y_{k}$). The observation noise should also follow a Gaussian distribution with zero mean and covariance *R* (~N(0,R)). Although *H* and *R* can vary at each time step *k*, it is common that they adopt a constant value.

In the update phase, the covariance is also updated to obtain the “a posteriori” estimate covariance $P_{k}$ according to Equation (30).
(30)$$P_{k}=P_{k}^{-}-K_{k}HP_{k}^{-}.$$

As a result of the update phase, the final breathing parameter ${\hat{x}}_{k}$ is estimated, which is the output of this algorithm. However, the Kalman filter is an iterative method that recalculates ${\hat{x}}_{k}$ at each time step. Therefore, the “a posteriori” state estimate ${\hat{x}}_{k}$ at the current time step will be the previous state estimate ${\hat{x}}_{k-1}$ at the next time step. The same happens with the covariance since the “a posteriori” estimate covariance $P_{k}$ at the current time step will be the previous estimate covariance $P_{k-1}$ at the next time step. In this way, the algorithm can start a new prediction process again (Figure 33). The whole process is repeated indefinitely. The output of the system at each transition is the “a posteriori” estimate of the respiration parameters ${\hat{x}}_{k}$.

#### 3.4.2. Results of the Analysis

Table 6 presents the results of the analysis for the wearable studies and Table 7 shows the results of the environmental studies. The second column of each table includes the specific data processing techniques used in each study. Figure 34 represents the number of works that use the different processing methods for wearable and environmental studies. The category “custom algorithm” was added to refer to processing algorithms that cannot be classified in any other group, as they are specific to the sensor used for respiration monitoring. It can be seen that “peak detection” in respiration signals and “frequency analysis” using the Fourier Transform were some of the most widely used methods by both wearable and environmental studies. The sum of the percentages of use of techniques based on the detection of levels in their different forms (peaks, maximum and minimum values, zero crossings, or thresholds) was 42% for the wearable category and 33% for the environmental systems. In the environmental category, the variability in data processing methods was much greater than in the wearable category, as a large number of studies applied “custom algorithms”. The use of wavelet decomposition or the Kalman filter was residual [102,131,165,166,193,212,232].

Figure 35 shows the figures of merit used to provide a value of sensor performance for the wearable and environmental studies. The categories “graphical comparison” and “graphical monitoring”, which could be considered as informal metrics, were added to the list of evaluation metrics of Section 3.4.1. The category “graphical comparison” refers to studies that visually compared the performance of the sensing system under evaluation with a reference system, but did not use an objective metric. The category “graphical monitoring” indicates that measurements from sensors were plotted, but no formal metric was calculated and no quantitative comparison was made. Figure 35 shows that “absolute error”, “relative/percentage error”, “Bland-Altman plot”, and “correlation coefficient” were the preferred formal metrics for wearable and environmental systems. The use of “root mean square error” [48,52,95,107,115,117,147,164,170,171,187,198], “linear regression” [68,161,167,183,209], and “accuracy” [7,76,133,161,179,190,193,207,216] was limited. Furthermore, the percentage of studies that provided an “informal” figure of merit was much higher for the wearable category (45%) than for the environmental group (17%). Therefore, a stronger assessment can be seen in environmental studies. In general, validation results show low error values and a high correlation with reference devices. The details for the different studies are included in the fourth column of Table 5 and Table 6. Fifty-two% of the studies that used relative or percentage errors provided a value less than 5%, while only 12% reported a value greater than 10%. Correlation coefficients greater than 0.95 were provided by 46% of the studies [19,52,61,112,126,149,176,180,208,215,222,228,232]. Regarding absolute error, 78% of the studies that calculated the RR as the breathing parameter provided a value less than 2 bpm [5,19,64,115,124,132,135,172,175,176,177,182,189,195,201,207,218,224]. No study reported an absolute error value greater than 4 bpm. In relation to the Bland-Altman analysis, the mean of the differences was less than 0.2 bpm in 49% of the studies that provided data on this metric [48,66,68,69,78,94,95,112,115,126,167,170,172,180,195,200,210,228].

Regarding the tools for measurement processing, Figure 36 shows the distribution of use of the different tools for the wearable and environmental respiration monitoring systems. MATLAB was the preferred software for both types of systems, since it was adopted by half of the studies, while NI Labview was the second most used tool as it appeared in 20–30% of the works [2,3,19,63,71,85,86,87,88,127,129,131,138,142,146,153,163,201,218,219,220,223,225]. The rest of the tools relied heavily on the specific sensor used to capture the data. For instance, Audacity, as a sound processing tool, could only be used in microphone-based respiration monitoring [162]; OpenCV, as a computer vision library, was suitable for respiration monitoring through images [187,216]. Therefore, the use of the rest of the tools was residual.

## 4. Discussion

Respiratory monitoring has been actively investigated in recent years, as can be deduced from the high number of studies included in this systematic review of the literature. While monitoring breathing in hospital or controlled environments poses fewer problems, the main research challenge is to monitor breathing for a long period in the user’s daily environment.

Following the approaches of previous works [24], two different sets of systems for respiratory monitoring were identified. On the one hand, wearable systems have the advantages that they can be used in any environment, either indoors or outdoors, are generally easy to install and, in most cases, inexpensive. However, the level of obtrusiveness can determine the acceptability of this type of systems. Some sensors, such as those designed to be worn on the face or neck, are more obtrusive. A set of wearable sensing technologies are less invasive. This may be the case of those that are worn in the chest, abdomen, arms, or wrist. This might be one of the reasons why the detection of chest movements is the predominant approach. Another reason could be that chest seems to be the part of the body that presents the greatest variations in its state as a result of the respiratory activity. However, most of these technologies require users to wear a belt on the chest or abdomen, electrodes that make contact with the skin, or tight clothing to detect the movement of the thorax [254]. These restrictions might cause discomfort in the long term or, in extreme cases, even skin problems. The proposal of Teichmann et al. [56] is original since the sensor is carried in a pocket of a shirt that does not need to be tight. This represents an advantage over other approaches, although some users may find the lack of integration into clothing uncomfortable. Future research can go in that direction. A common drawback of wearable systems is that they are heavily affected by artifacts caused by non-breathing movements. This leads to larger measurement errors, which can even compromise the viability of the sensing systems in extreme cases. On the other hand, environmental sensors have the advantages that they are non-invasive, data transmission can be done with cable communication and battery life does not limit their operation. However, their scope is restricted to a particular area. Any change in the environment (for example, the relocation of furniture) can modify the detection capabilities. Additionally, some technologies, such as computer vision, present privacy concerns, which may affect user acceptance [271]. Environmental sensors seem suitable for home or hospital applications, but not for continuous monitoring of moving subjects. In fact, usability is a big challenge in respiratory monitoring. Several authors have highlighted this fact [10,19,51,52,115,159,187,254]. However, despite this, we identified a clear gap in the literature since it was not possible to find any usability analysis of the sensors implemented in the existing studies. Future research should also focus on usability. For that, well-known usability tests can be applied to evaluate the level of acceptance of technology by its potential users. For example, the User Acceptance of Information Technology (UTAUT) model [272,273] may serve. This model was applied previously to evaluate smart wearable devices [274], including health care [275,276], and m-health devices [277]. Other parameters, such as size or weight, can also affect the adoption of the technology. These parameters have only been provided in a limited number of studies [2,3,17,21,49,62,64,68,72,85,93,94,97,98,99,103,105,108,110,113,114,116,119,120,124,126,135,136,138,141,143,144,146,148,161]. Future works should also consider size and weight as important factors in the design of sensing systems, subjecting them to evaluation.

Regarding the type of sensors used for respiratory monitoring, fiber optic sensors prevailed in the wearable category. This may be due to their insensitivity to electromagnetic fields, their high resistance to water and corrosion, and their compact size and low weight [125]. This technology also allows monitoring different types of physiological parameters simultaneously [278]. In addition, resistive sensors and accelerometers were the second and third most widely used technologies. This might be explained as they are suitable for detecting movement variations, and their design is simpler than other technologies, such as capacitive, pyroelectric, or piezoelectric sensors, among others. In relation to the environmental category, most researchers designed radar-based sensors. Cameras were also widely used. The great development of computer vision technology in recent years [279] makes the detection of chest movements through video image technically feasible. However, cameras present privacy concerns, which may be why radar sensors are the preferred non-contact technology. Radar systems are also small in size, low cost, and simple in structure, which provides advantages in installation and operation [280]. The researches that decided to integrate the sensors into everyday objects again opted for fiber optic technology and sensors based on the measurement of resistance changes. This could be due to the advantages of these technologies, which have been mentioned before. However, the use of non-object-embedded environmental systems was the predominant approach, as they do not require users to be in permanent contact with an object, increasing system applications.

Comparing the performance of sensors is a challenge. It is difficult to compare the performance of different studies, since there is no standardized test to validate the sensors. Authors designed customized experiments with great differences among them. Many aspects were defined differently: the type and values of the respiratory parameters considered, the positions of the subjects during the tests, the number of human participants involved in the experiments, their characteristics, or the duration of the experiments. Differences were also found in the inclusion of motion artifacts and in the mechanical devices that simulated respiration, among others. A consequence of this is that performances provided by existing studies are not comparable, since they were obtained under different test conditions. Therefore, a future research effort is to design a common evaluation framework. This framework should include quiet and rapid breathing and different postures, such as standing, lying, or sitting. Experiments should include motion artifacts since they affect sensor performance, as shown by several studies [53,108,157,158]. Additionally, they should involve a number of users high enough to obtain significant results. In general, the number of subjects participating in existing studies remains low.

In view of the results shown in Section 3.3.2, it is a fact that existing studies carry out short experiments to validate the sensors. However, little attention is paid to their long-term behavior. The effect of temperature, sensor aging, or the characteristics of the carrier subjects (such as height or weight) on the sensing systems have not been actively explored. A sensor that works well in a laboratory environment might not work as well in a real setting when used for a long time. If there were errors in the measurements, this would cause frequent recalibrations of the sensors. Therefore, a research challenge is to test the behavior of respiratory monitoring systems in more realistic environments. The declared performances in laboratory or controlled settings are high. The challenge is to prove that they are equally high in real-world usage. Sensor aging might be a problem in terms of system performance. However, it is less critical in terms of cost as replacement of sensing parts is generally affordable due to its typically low price.

Regarding the declared performances, the validation of the sensors should be done considering reference devices. This is the approach adopted by most of the studies. Other validation methods, such as the use of metronomes or artificial prototypes, are less common since, unlike reference devices, they are not well-established systems that can be acquired by the scientific community to replicate experiments or compare results. There is less consensus with respect to the figure of merit used to determine the accuracy of the sensing systems. The relative, absolute and RMS errors, the slope of the linear regression line, and the correlation factor have been considered. It is also common to apply the Bland-Altman analysis [281]. It not only provides information on the differences between the measurements of both sensors but also shows the variation of the differences with respect to the magnitude of the measurements. In addition, the standard deviation of the difference is also used to obtain the upper and lower limits of agreement. The high variability of the figures of merit makes it difficult to compare the studies. One issue of respiratory monitoring is that the acceptable margin of error is not clearly defined and, therefore, it is difficult to determine whether a new sensing system is in agreement with reference devices. This may be a consequence of the lack of a common experimental framework, since the margin of error depends on the specific experiments carried out. For example, the acceptable error may be different for slow or rapid breathing. Ideally, both the mean of differences and the limits of agreement in the Bland-Altman analysis should be provided. As complementary information, it would also be interesting to have the mean absolute or relative errors, or the correlation factor. This would facilitate comparison of system performance among studies. However, this is not the most common approach and only a limited number of studies have incorporated it [10,44,57,60,61,65,78,92,100,109,112,114,122,123,124,132,163,165,166,167,168,169,176,178,191,196,205,206,211,224,228]. 

Additionally, the parameter to be sensed varies among studies. The most common breathing parameter obtained by existing studies is the RR. However, several studies calculate volume parameters, which are useful for many applications. There are studies that provide both [2,49,52,61,116,122,147], although the calculation of the volume parameters is a challenge since it depends on the specific technology. An approximation for a capacitive textile force sensor can be found in the work of Hoffmann et al. [17]. However, this is still and open research topic. Obtaining an accurate estimate of volume parameters using the sensing techniques presented in this review is not easy, especially for wearable systems.

Regarding processing algorithms, it can be concluded that detection of peaks, maximum and minimum values, thresholds, or zero crossings were effective in determining respiration parameters. Frequency analysis also provided good results. This aspect seems to have been successfully resolved in existing studies. The use of other processing techniques is residual, as they generally require more computing resources, are more complex, and are highly dependent on the specific design of the sensing system.

Wired and wireless transmission are used equally, as the type of transmission is usually determined by the type of sensor. Within wireless systems, Bluetooth was the preferred option. This may be explained because most sensing systems communicate with a smartphone/tablet or PC that is close to the sensing unit. In fact, PC processing is the main trend. This can be a consequence of the majority of studies presenting laboratory prototypes that are far from usable portable systems. In general, authors do not give much thought to the amount of resources that the processing algorithms use as they perform centralized offline processing on a PC using numerical computing software, such as MATLAB. This could compromise the real time operation of the systems when they are running continuously. Future research efforts should focus on designing suitable processing techniques to run ubiquitously in real time on the same microprocessor of the sensing unit or on smartphones.

Power consumption is crucial in wearable respiratory monitoring. Most studies did not provide information on power consumption or battery life. In addition, there was no consensus on the measurement procedure and the energy parameters that should be provided. In this context, it is very difficult to compare the power consumption provided by the different studies fairly. For example, battery life varies greatly depending on factors, such as data transmission procedure (continuous, intermittent, or without transmission), sensor operating time (non-stop, several hours a day, etc.), monitoring visualization (real-time display, without visualization, etc.), or the inclusion of the processing of the measurements in the power study, among others. When a study provides the battery life of a respiratory sensor, not only must the capacity of the particular battery used in the study be provided, but also other characteristics that affect its performance: depth of discharge, cycle lifetime, or c-rate [282]. It would also be interesting that researchers indicate the power consumption of each component of the system and not just the total autonomy of the device [98]. This would allow the identification of the critical components and facilitate the comparison of different systems. Given that autonomy is a limiting factor, strategies to reduce power consumption are required [135]. However, only a few works have implemented them. Several respiratory monitoring studies focused on transmission technology, since it is usually the most demanding module [91,98]. For example, technologies, such as Wi-Fi or Bluetooth, consume more energy than Impulse Radio Ultra-wideband. Other strategies adopted were the down-sampling of data to reduce computational load [119]. The limited number of works that adopted energy saving strategies may be a symptom that researchers are more focused on validating their sensors than in real-world applications. This can also be inferred from the short battery life indicated by most studies that include this data (less than 12 h typically). Furthermore, the use of energy harvesting techniques in respiratory monitoring is another open research question, since the number of studies that implemented them is still residual [77,84,104,135,240,241,242,243,244,245,246,247,248,250,251,252,253]. Most studies presented laboratory experiments instead of functional prototypes. 

An ideal breathing sensor should be mobile, easy to use, imperceptible and immune to body motion [48,119,135]. Several authors agree that a system that covers all these aspect should be successfully integrated into clothing [21,59,65,69,84,85,94,103,108,113,119,123,142,143,151]. This is a consequence of the fact that long-term home monitoring entails direct connection between patient and system. However, this poses several problems, such as the adaptation of the sensing system to different sizes of clothing, the integration of the energy supply, or the washing of sensors, among others. In fact, this is an open research question.

## 5. Conclusions

This paper presents a systematic review on sensors for respiratory monitoring, filling a gap in the state of the art since no published reviews analyzing respiratory sensors from a comprehensive point of view could be found to the best of our knowledge. As a result of several searches, an overwhelming number of studies was found. They were sorted by relevance and, finally, 198 studies were obtained to be examined in detail. They were classified into two groups: wearable and environmental sensors. Several aspects were analyzed: sensing techniques, sensors, breathing parameters, sensor location and size, general system setups, communication protocols, processing stations, energy autonomy, sensor validation experiments, processing algorithms, performance evaluation, and software used for the analysis. As a result, detection of chest movements was identified as the most common technique using fiber optic sensors for the wearable systems and radar sensors for the environmental systems. The RR was the most common breathing parameter obtained in 68% of the studies. Most of the studies performed centralized measurement processing on a PC using MATLAB software. Bluetooth was by far the prevalent communication technology (60% of the wearable studies adopted it), and almost all wireless respiration sensing systems were battery powered. The most common validation approach was to use a reference device to perform real tests on real subjects. Furthermore, a high percentage of studies obtained the breathing parameter after performing frequency analysis or peak detection on the measurements. Meanwhile, the most common figures of merit selected to provide evidence on sensor performance were absolute and relative errors, Bland-Altman analysis, and correlation coefficients.

This review also identified future research challenges. One of them is the need to define a common framework to validate the sensors, since each author carried out his or her own experiments. This makes it difficult to compare sensor performances. Similarly, measurements of power consumption were made under different conditions. A common measurement procedure is required to compare sensor autonomies fairly. There are no long-term evaluations that study the effect of aging, environmental conditions, or characteristics of the subjects on sensor performance.

Usability tests are also lacking in existing studies. Similarly, the figure of merit to provide sensor performance varies from one study to another. The Bland-Altman analysis was identified as the most appropriate method to validate the sensors against reference devices. Other research challenges are the implementation of energy-saving or energy-harvesting strategies, the application of respiratory sensors to real-world scenarios, or the calculation of volume parameters in the different sensing technologies. All these are remaining research efforts.

Finally, several authors highlighted the integration of respiratory monitoring sensors in clothing as a promising technology. This is a future research effort, which presents several challenges for a feasible, long-term, and unobtrusive solution.

## Figures and Tables

**Figure 1 sensors-20-05446-f001:**
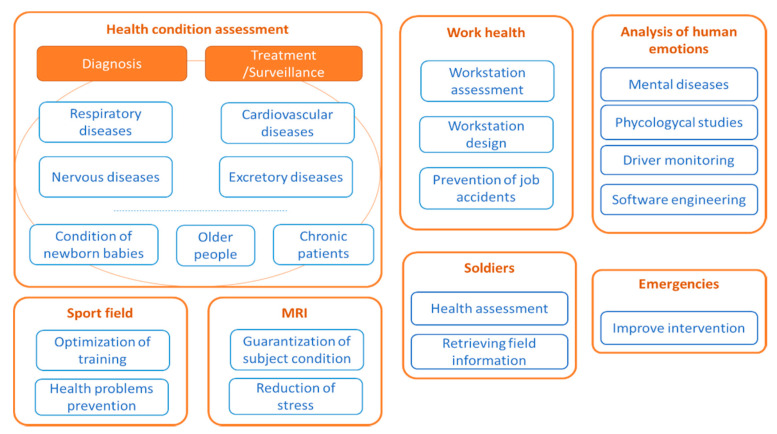
Most common application fields of sensing systems to monitor breathing.

**Figure 2 sensors-20-05446-f002:**
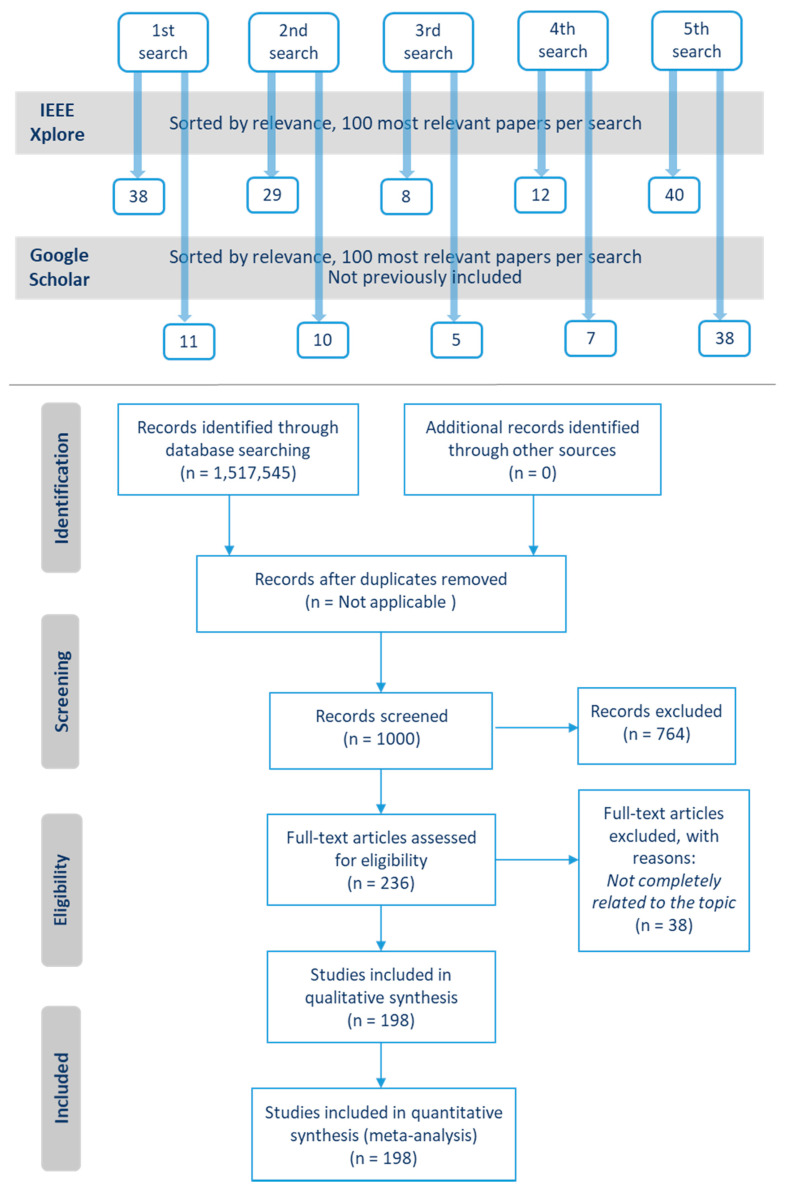
Literature search results and selection procedure (**top**). PRISMA diagram (**bottom**).

**Figure 3 sensors-20-05446-f003:**
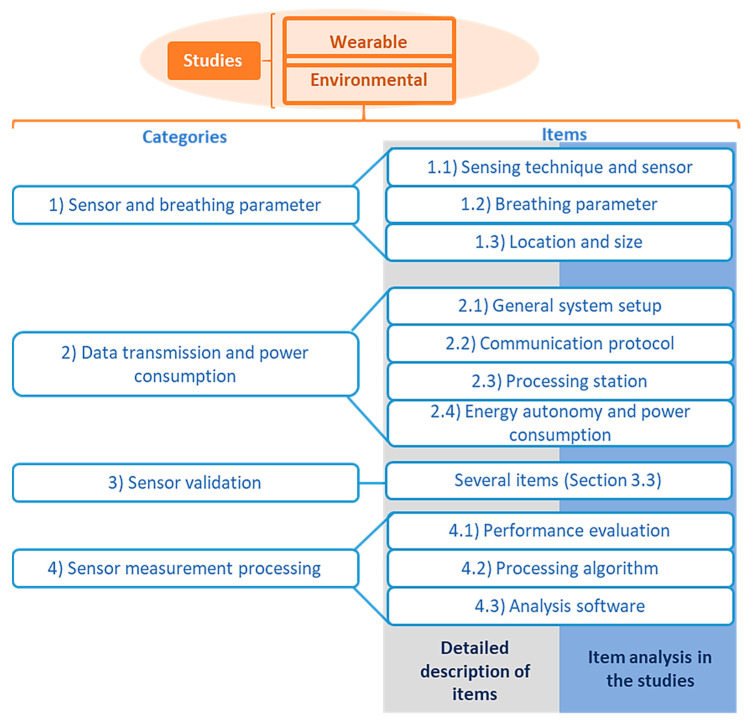
Analysis structure.

**Figure 4 sensors-20-05446-f004:**
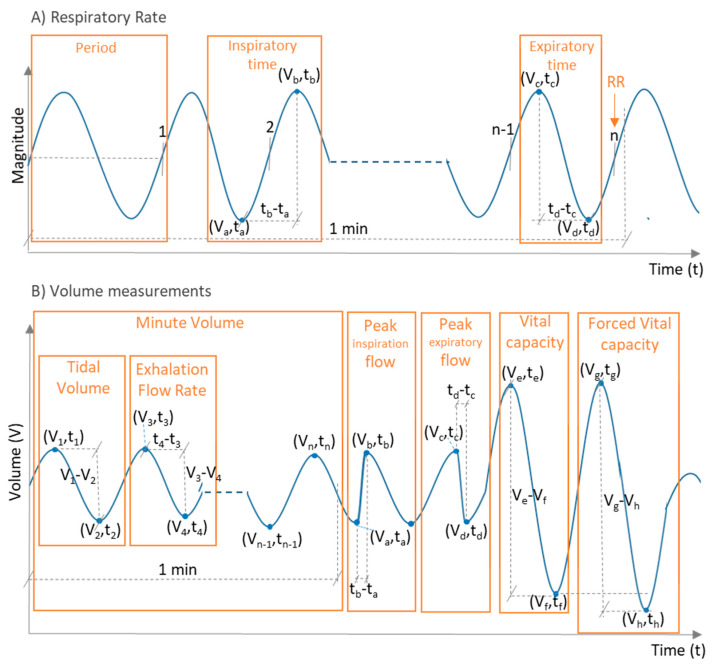
Graphical explanation of the different breathing parameters. Signal (**A**) could come directly from the ADC (analog-to-digital converter) of the sensing system, although it is also possible that it represents physical respiration magnitudes. This figure shows a general representation that is not contextualized to a specific sensing system. The same goes for signal (**B**).

**Figure 5 sensors-20-05446-f005:**
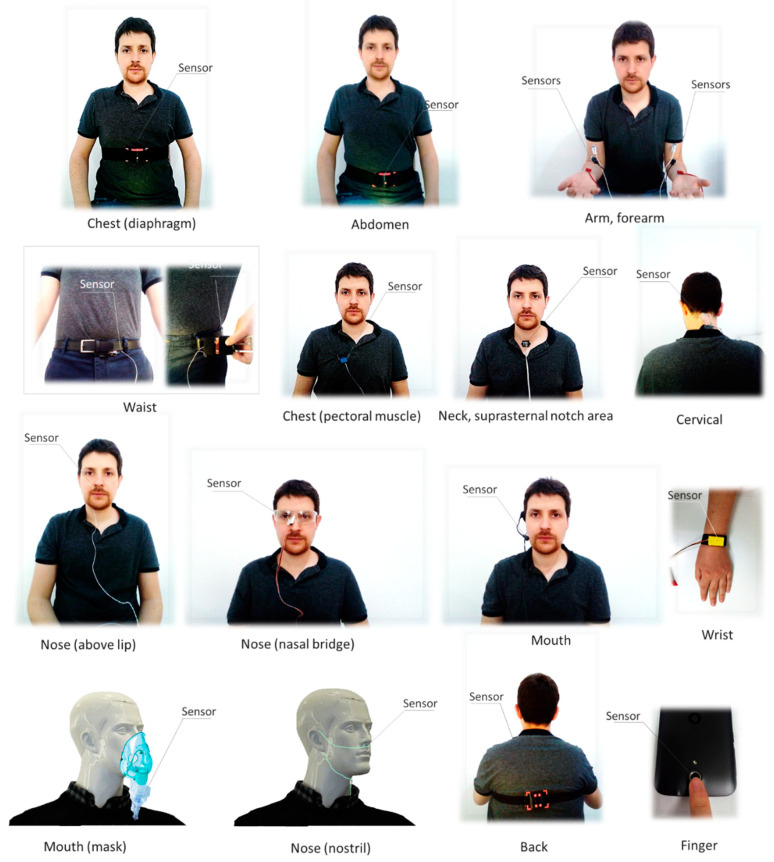
Most common sensor locations for respiration monitoring. The sensors shown are for contextualization purposes.

**Figure 6 sensors-20-05446-f006:**
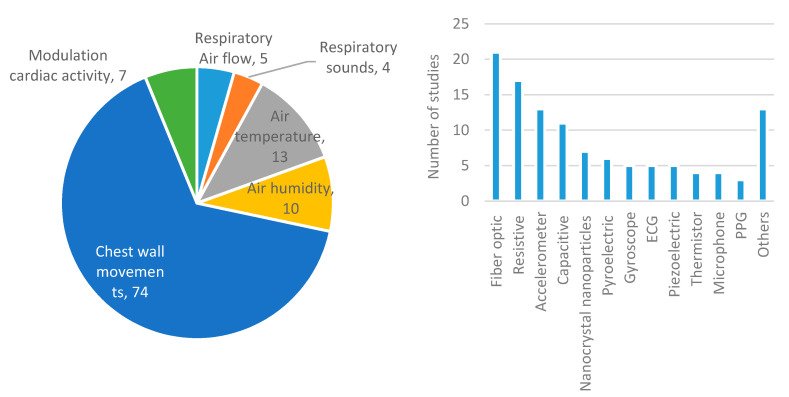
Distribution of sensing techniques (**left**) and sensors (**right**) used in the studies of the wearable category.

**Figure 7 sensors-20-05446-f007:**
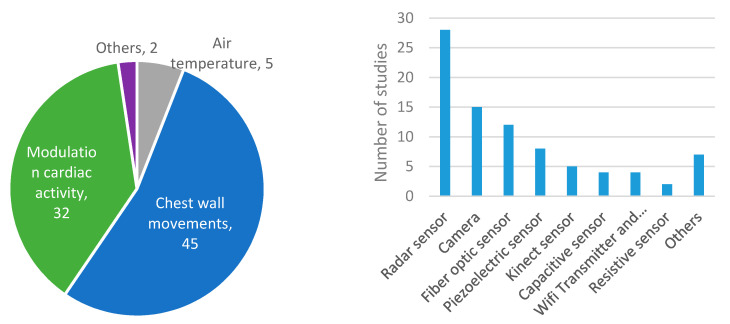
Distribution of sensing techniques (**left**) and sensors (**right**) used in the studies of the environmental category.

**Figure 8 sensors-20-05446-f008:**
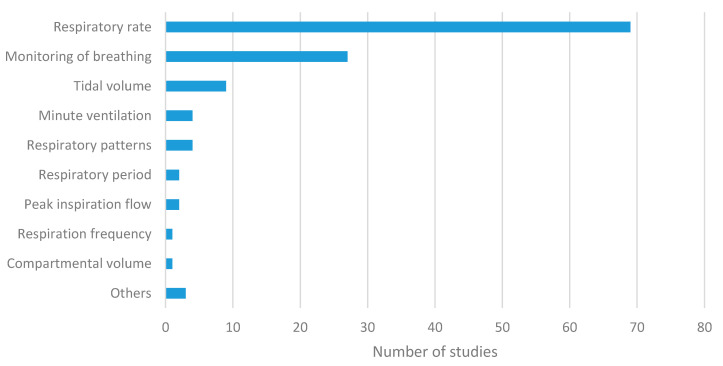
Number of studies obtaining the different respiratory parameters for the wearable (**top**) and environmental (**bottom**) categories.

**Figure 9 sensors-20-05446-f009:**
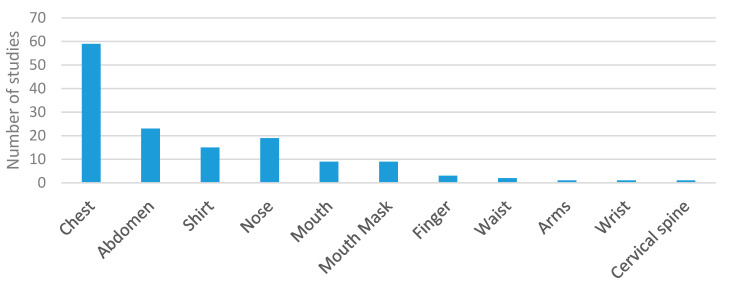
Distribution of sensor location for the wearable studies.

**Figure 10 sensors-20-05446-f010:**
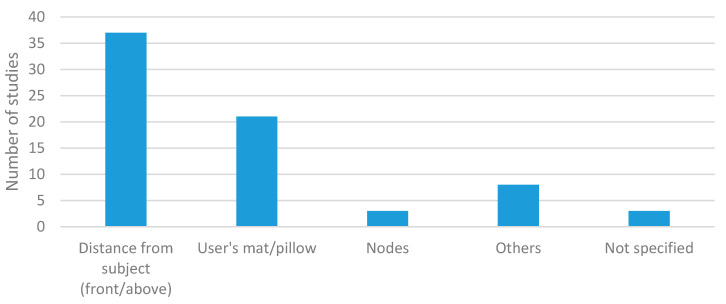
Distribution of sensor location for the environmental studies.

**Figure 11 sensors-20-05446-f011:**
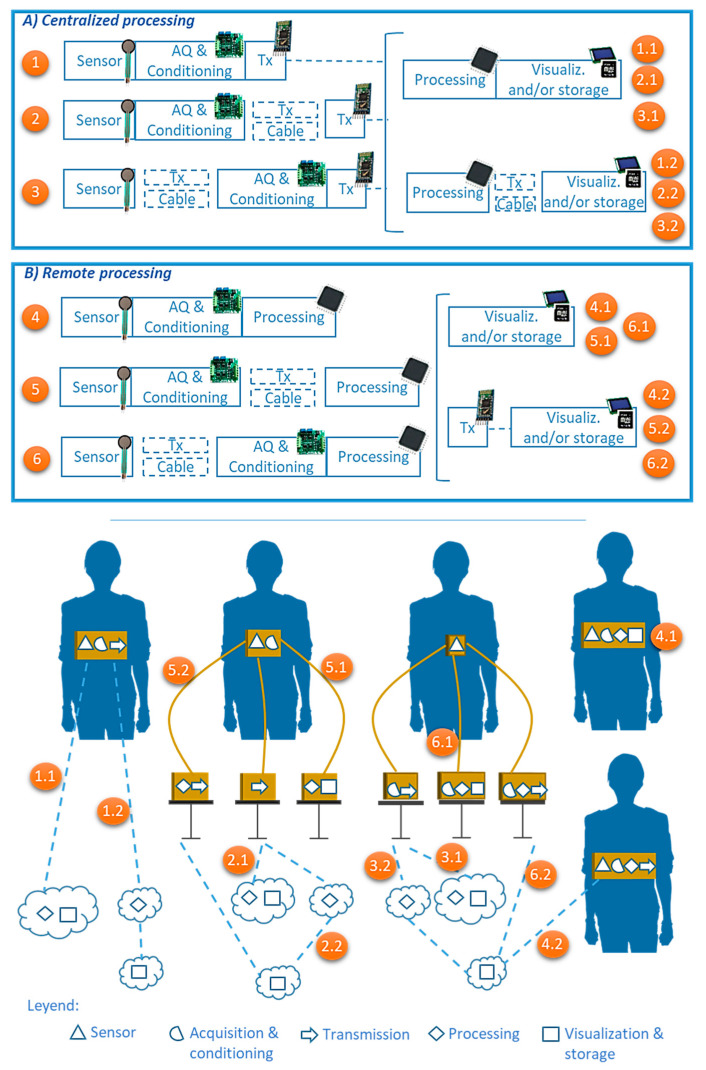
Representation of possible setups of respiratory sensing systems. (**A**) perform data processing on a centralized processing platform and (**B**) perform data processing near the remote sensing unit.

**Figure 12 sensors-20-05446-f012:**
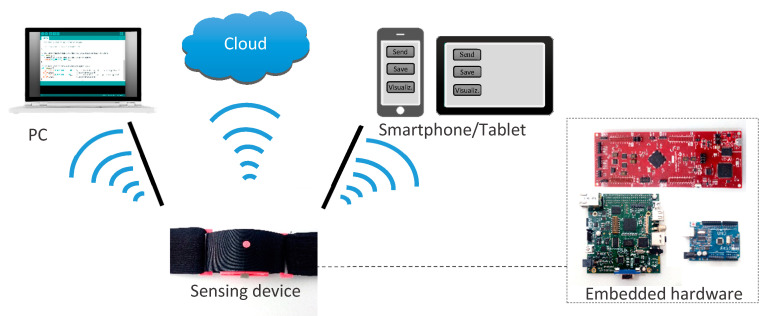
Representation of possible setups of respiratory sensing systems.

**Figure 13 sensors-20-05446-f013:**
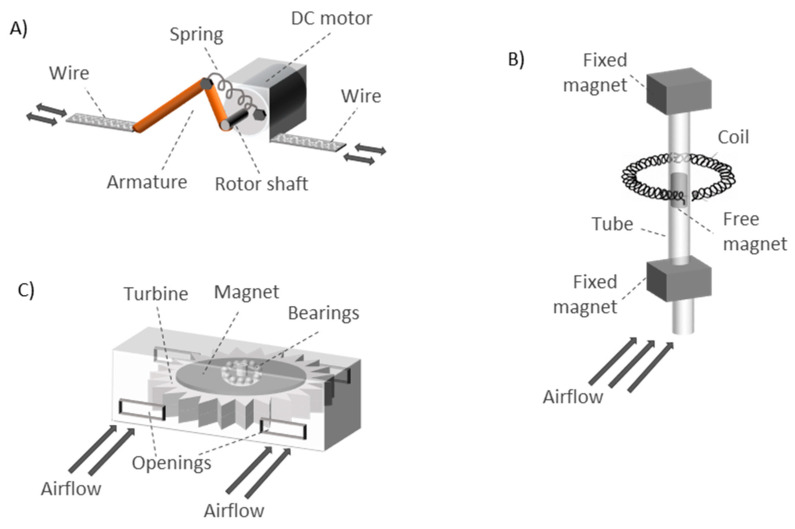
Schemes of energy harvesting using magnetic induction generation: (**A**) DC generator activated by chest movements (figure inspired by Reference [135]), (**B**) tube with fixed and free magnets moved by airflow (figure inspired by Reference [240]), and (**C**) turbine moved by airflow (figure inspired by Reference [241]).

**Figure 14 sensors-20-05446-f014:**
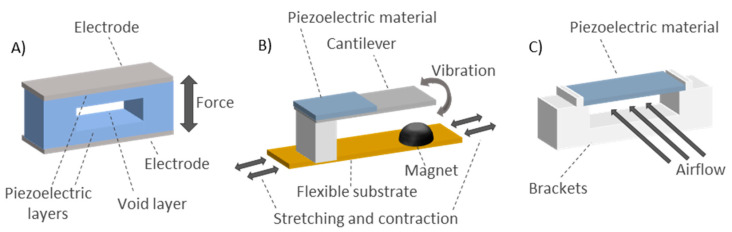
Piezoelectric energy harvesters. Three possible configurations are shown: (**A**) power generation based on compression or stretching movements associated with breathing (figure inspired by Reference [244]), (**B**) energy harvesting based on vibration amplified by a magnet (figure inspired by Reference [243]), and (**C**) technique using low speed airflow (figure inspired by Reference [245]).

**Figure 15 sensors-20-05446-f015:**
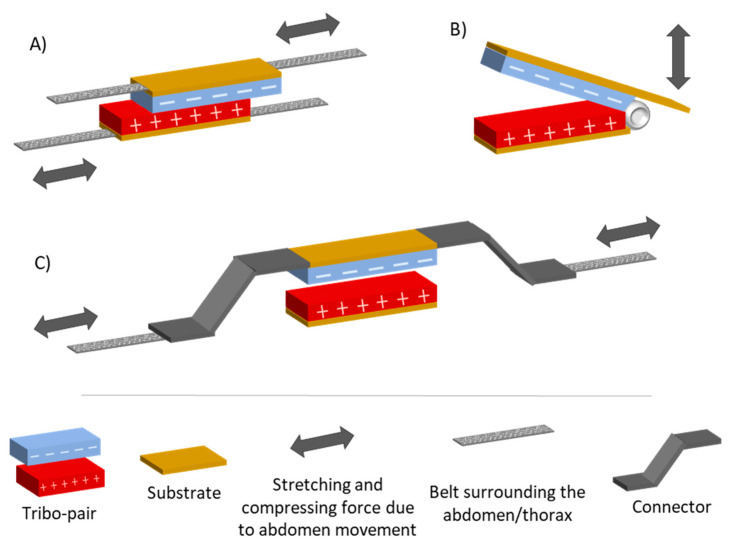
Setups for triboelectric energy harvesting. Three possible configurations are shown: (**A**) flat belt-attached setup (figure inspired by Reference [246]), (**B**) Z-shaped connector (figure inspired by Reference [77]), and (**C**) movable and fixed supports (figure inspired by Reference [247]).

**Figure 16 sensors-20-05446-f016:**
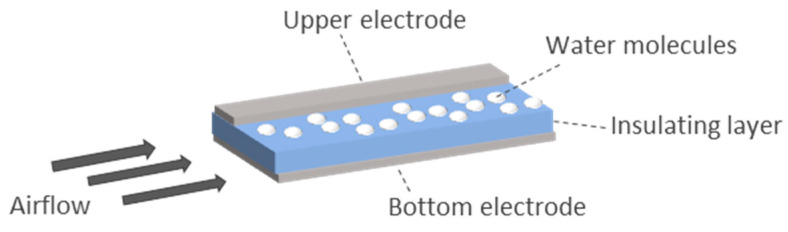
Electrostatic energy harvesting based on the variation of the area of the upper electrode owing to humidity of the exhaled air (figure inspired by Reference [248]).

**Figure 17 sensors-20-05446-f017:**
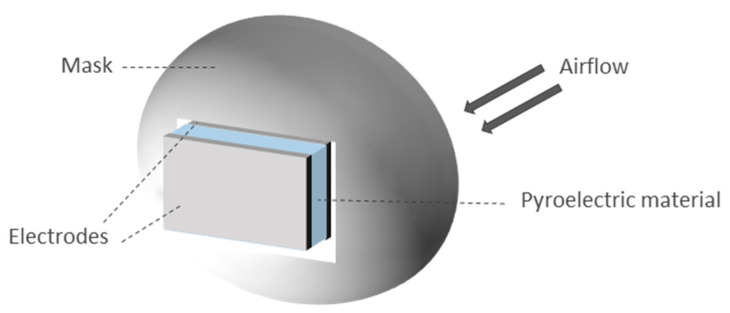
Schematic of a pyroelectric energy harvester using a mask-mounted breathing prototype (figure inspired by Reference [253]).

**Figure 18 sensors-20-05446-f018:**
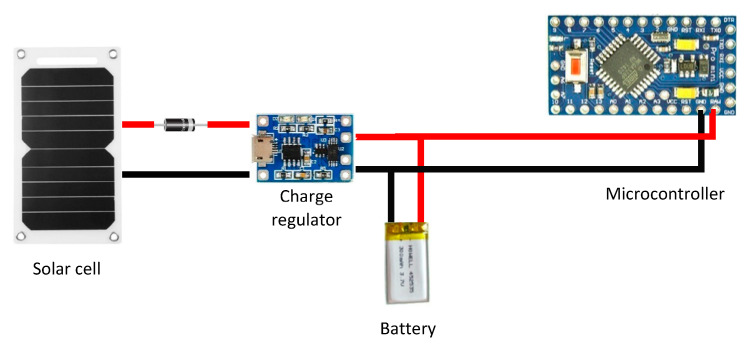
Example of a solar-powered system composed of a solar module, a charge regulator and a microcontroller. The voltage regulator receives an input voltage from the solar cell in the range of 0.3 V to 6 V. The charge regulator manages the charge of the battery (at constant voltage and current). The battery is connected in parallel to the internal voltage regulator of the microcontroller of the system.

**Figure 19 sensors-20-05446-f019:**
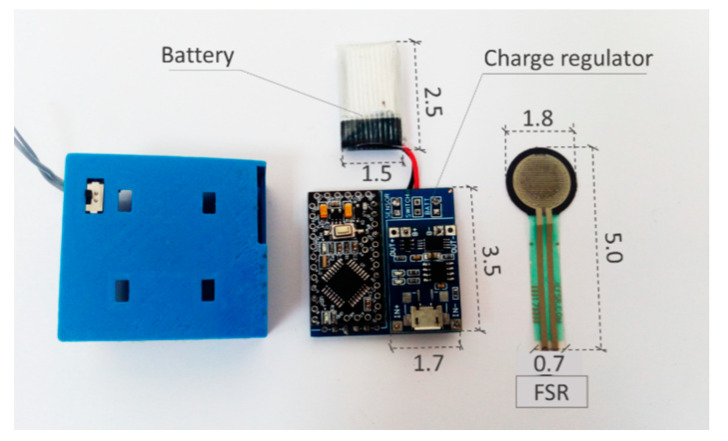
Charge regulator and battery (low capacity, 150 mAh) integrated into the sensing prototype developed by Vanegas et al. [254], slightly modified. The sensor used in that prototype (a force-sensitive resistor) is included separately for size comparison. Units: cm.

**Figure 20 sensors-20-05446-f020:**
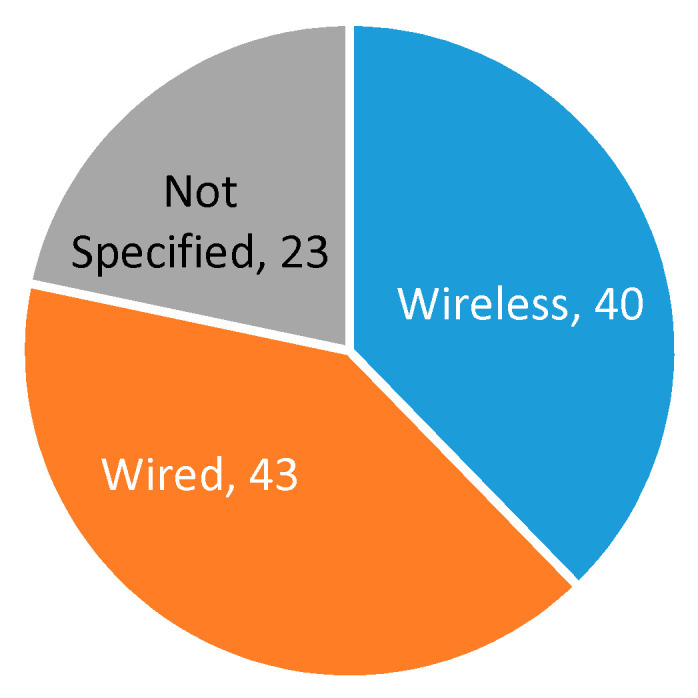
Number of studies adopting wired or wireless data transmission in respiration sensing systems.

**Figure 21 sensors-20-05446-f021:**
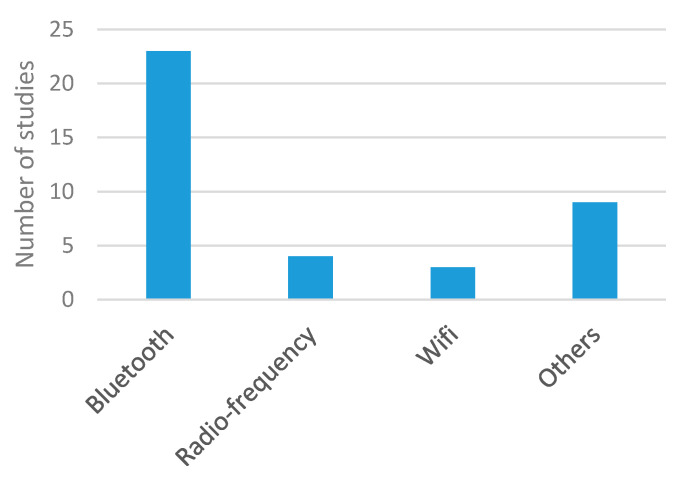
Number of respiratory monitoring studies that considered different types of communication technologies.

**Figure 22 sensors-20-05446-f022:**
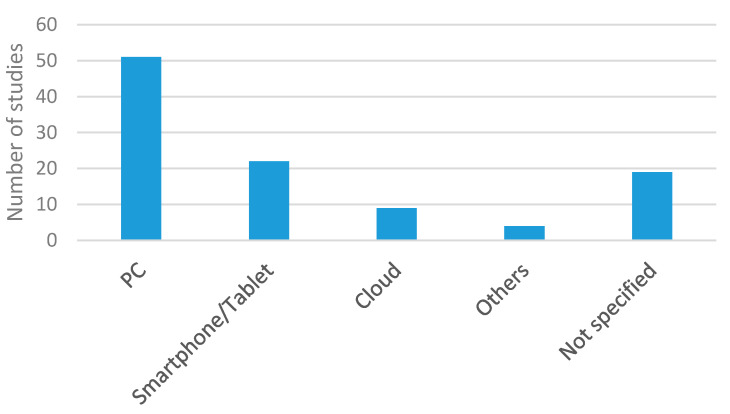
Number of studies adopting the different processing units.

**Figure 23 sensors-20-05446-f023:**
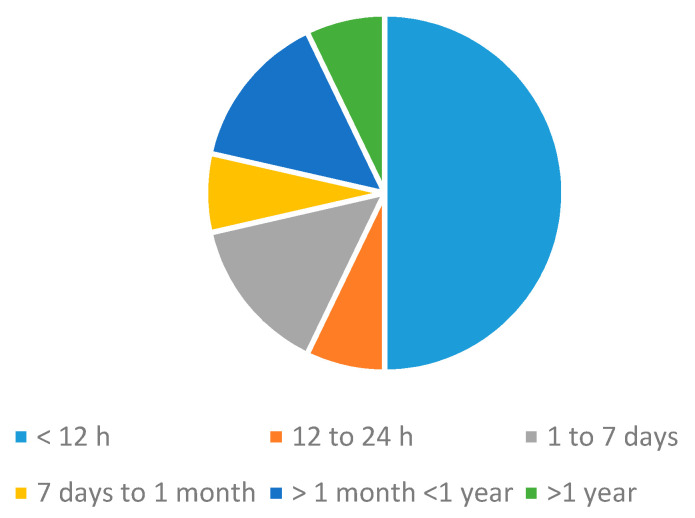
Distribution of battery lives reported in the respiratory monitoring studies.

**Figure 24 sensors-20-05446-f024:**
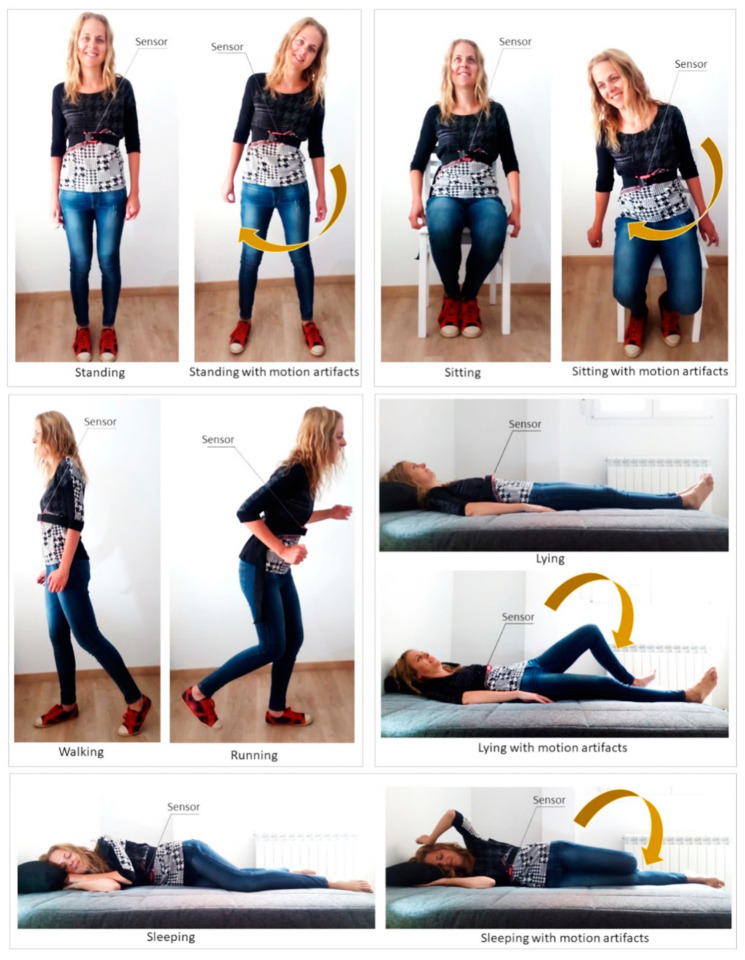
Common positions/activities to validate the breathing sensors (sitting, standing, lying down, walking, running, and sleeping). Chest sensor used as an example.

**Figure 25 sensors-20-05446-f025:**
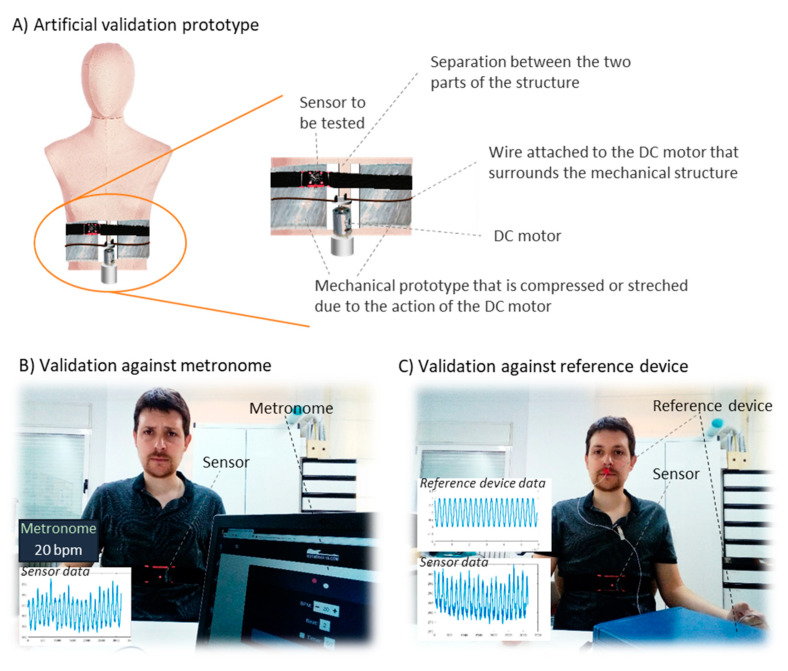
Representation of different validation approaches: (**A**) use of artificial validation prototypes, (**B**) validation using a metronome, and (**C**) validation using a reference device.

**Figure 26 sensors-20-05446-f026:**
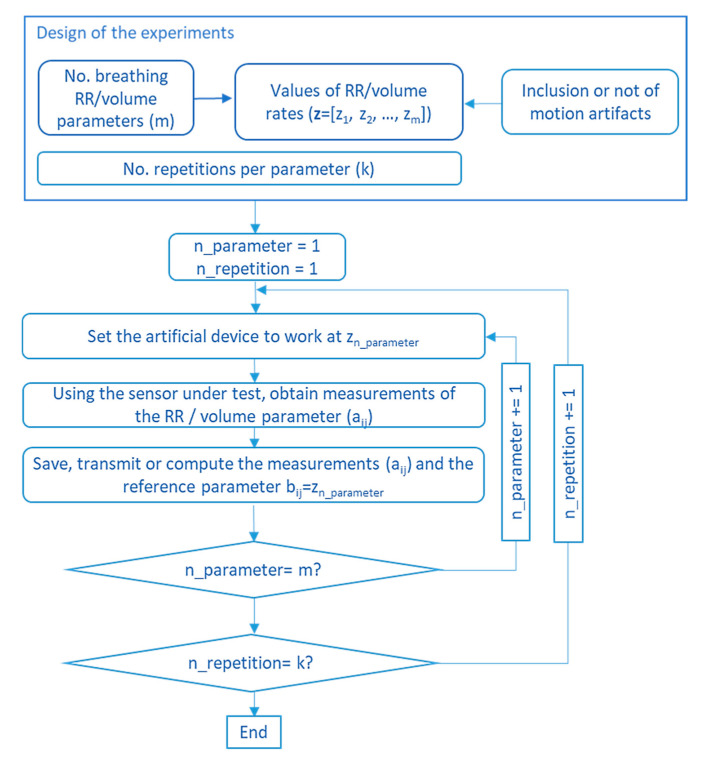
Flow diagram of a typical validation procedure using artificial prototypes.

**Figure 27 sensors-20-05446-f027:**
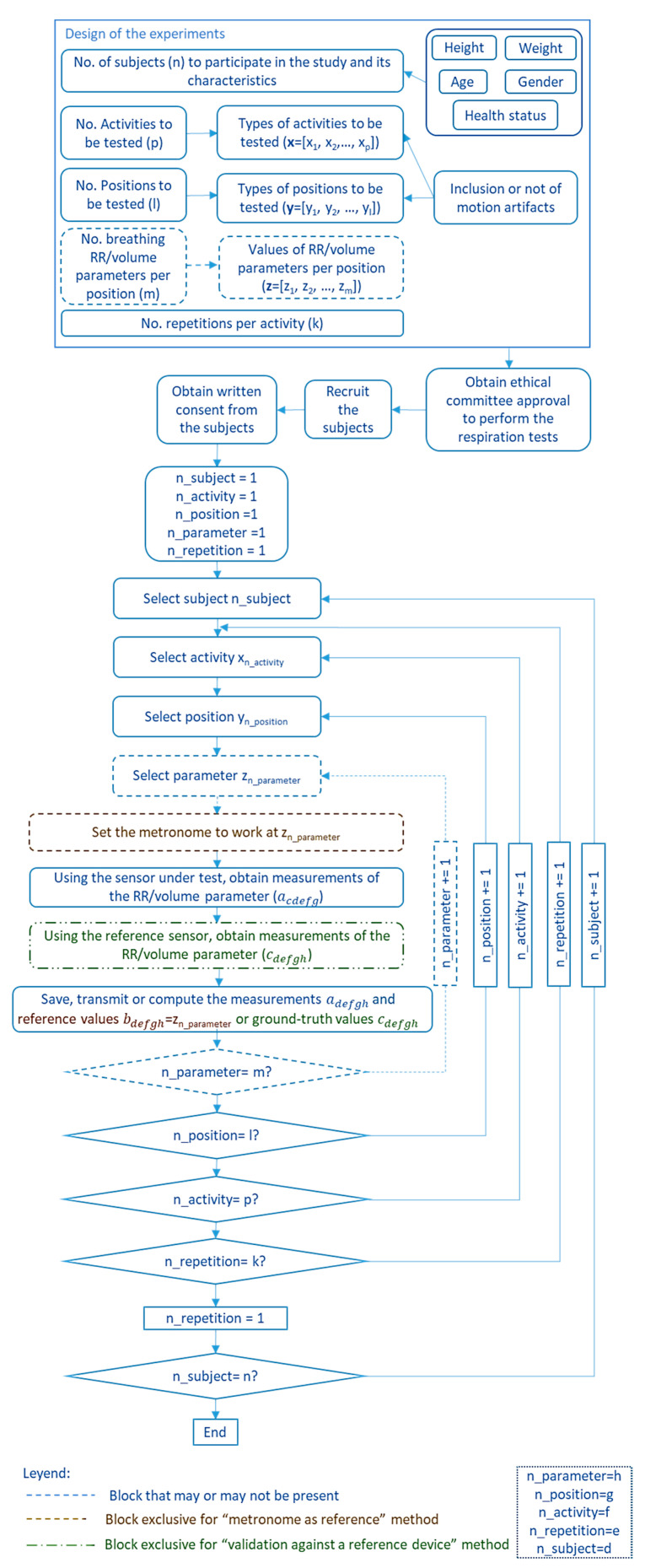
Flow chart for the validation of a respiration sensor using the methods “metronome as reference” and “validation against a reference device”.

**Figure 28 sensors-20-05446-f028:**
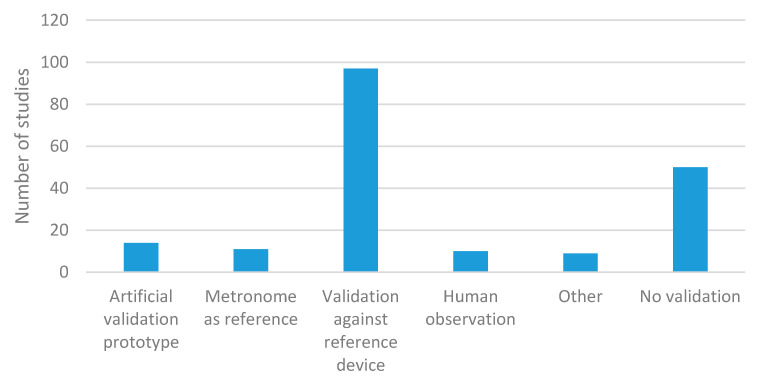
Number of studies that adopted the different validation approaches.

**Figure 29 sensors-20-05446-f029:**
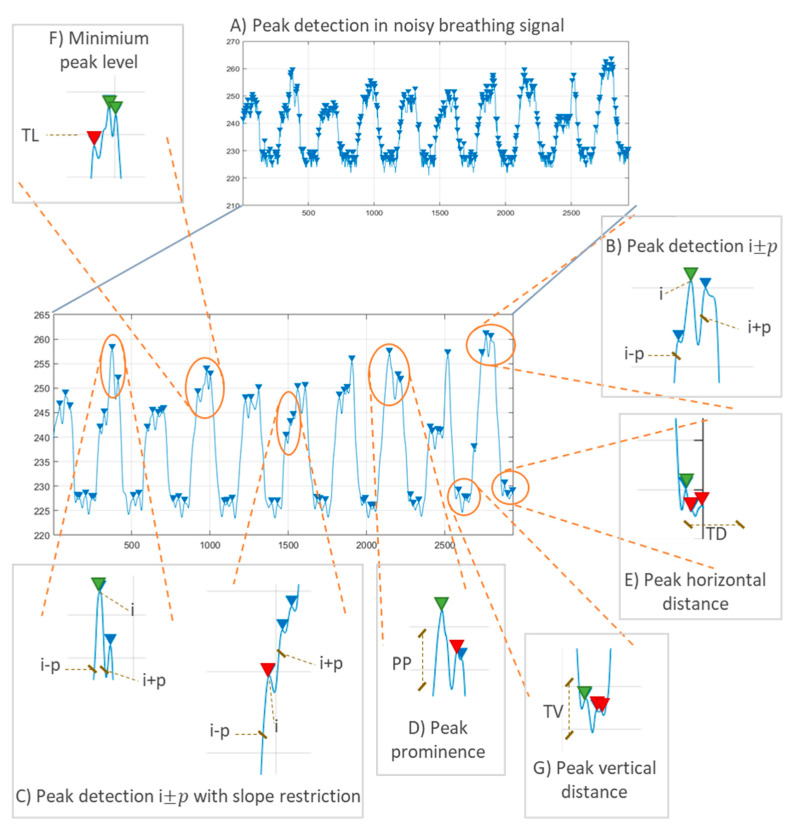
Peak detection of a sample respiration signal obtained from the public breathing dataset published in Reference [254]. (**A**) Peak detection of a noisy signal without filtering. (**B**) Peak detection imposing a restriction of *p* surrounding number of samples (in green the peak accepted). (**C**) Example of a peak accepted (left, green peak) and a peak discarded (right, red peak) when applying the slope restriction. (**D**) Example of a peak reaching (green) and not reaching (red) the minimum prominence level PP to be considered a valid peak. (**E**) Example of two peaks (red) not fulfilling the minimum horizontal distance restriction TD. (**F**) Example of a peak (red) not fulfilling the vertical minimum level restriction and two peaks that surpass level TL (green peaks). (**G**) Example of two peaks discarded (red) for not differing the imposed tidal volume (TV) level from a detected peak (green).

**Figure 30 sensors-20-05446-f030:**
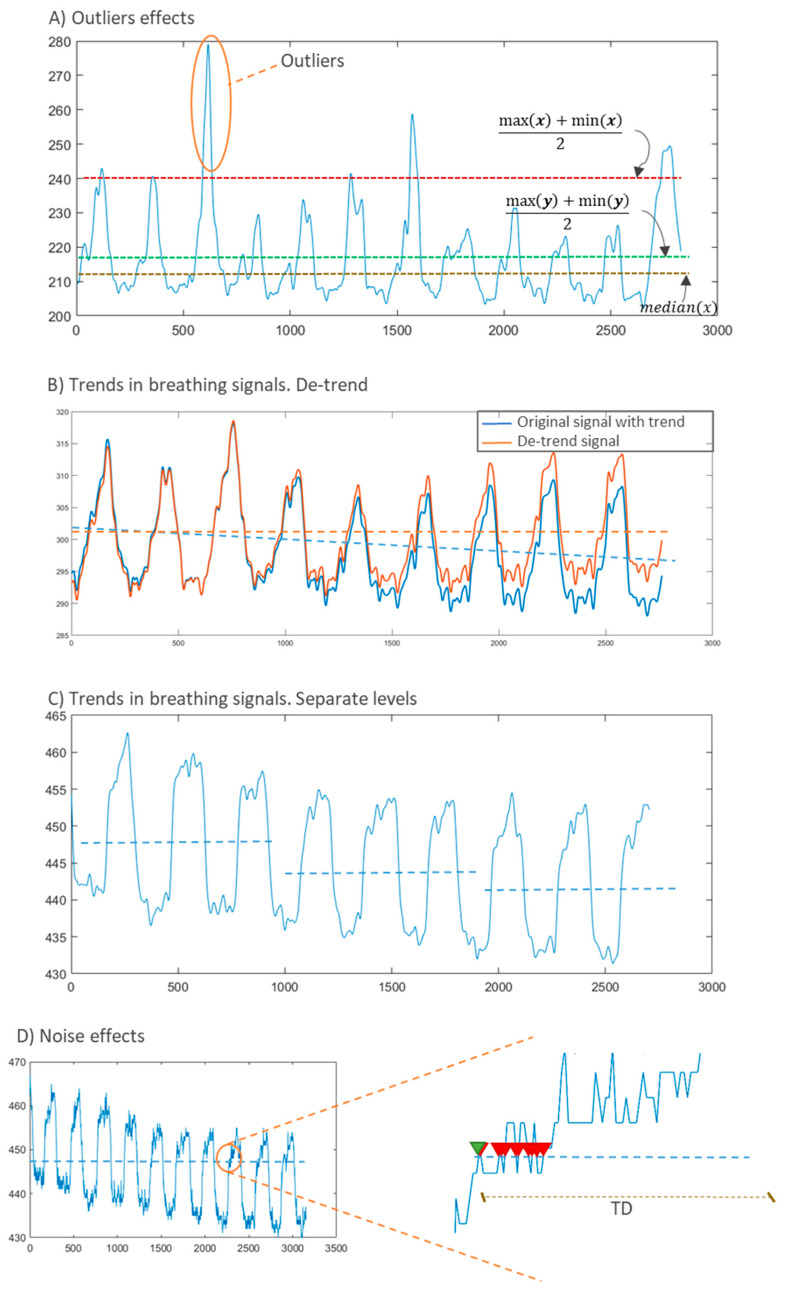
Zero-crossings method exemplified in a real signal obtain from the public breathing dataset of Vanegas et al. [254]. (**A**) Effect of the presence of outliers in the signals in the calculation of the “zero level”. (**B**) Example of a signal with trends and results of applying a de-trend processing. (**C**) Example of using different “zero levels” in a signal with trends. (**D**) Example of a noisy signal with several zero-crossings detected when only one of them (green) should have been considered.

**Figure 31 sensors-20-05446-f031:**
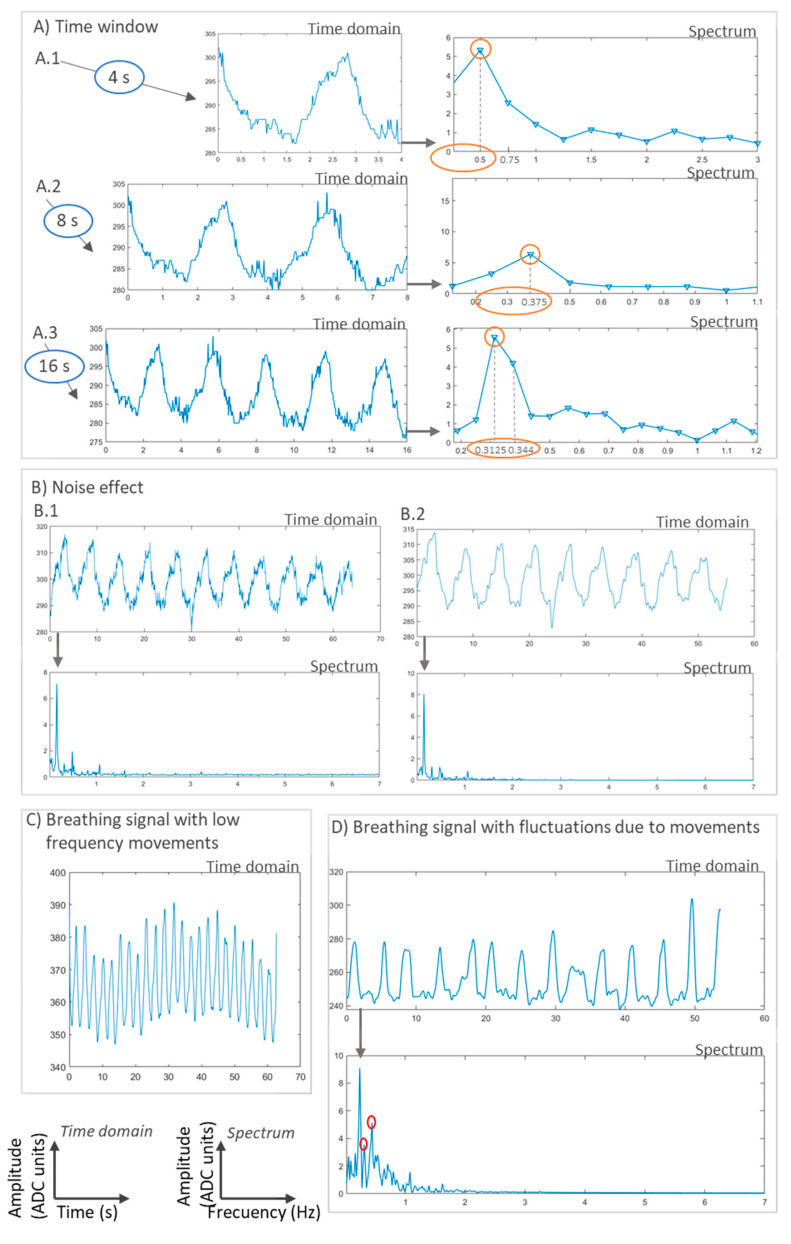
Frequency analysis of sample real respiratory signals obtained from a public dataset [254]. (**A**) Effect of the time window (4 s, 8 s, and 16 s) on the frequency calculation. The true frequency is 0.33 Hz (3 s period) and the sampling frequency is 50 Hz. Results for the 16-s time window (Table A3, 0.3125–0.344 Hz) are closer to the true value. (**B**) Effect of noise on frequency detection (noisy signal and its spectrum -B.1-, filtered signal and its spectrum -B.2-). (**C**) Example of a breathing signal with low frequency fluctuations. (**D**) Example of a breathing signal with fluctuations due to movements of the subject and its spectrum.

**Figure 32 sensors-20-05446-f032:**
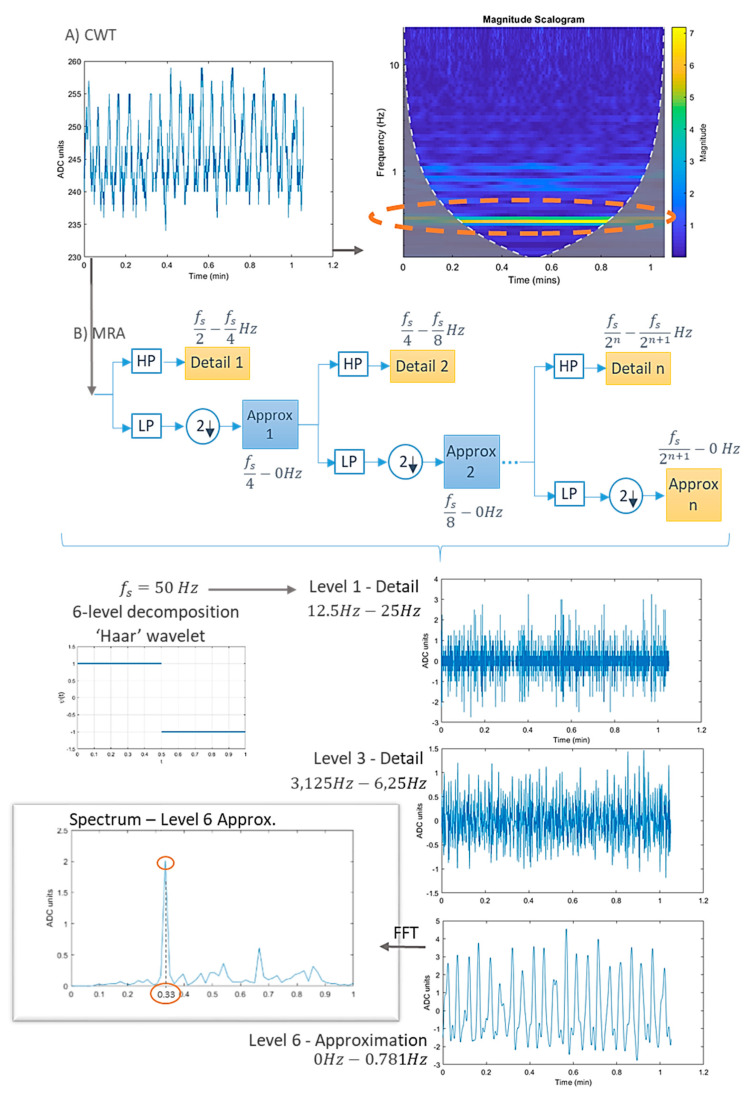
Wavelet transform. (**A**) 2D representation of the continuous wavelet transform (CWT) (**right**) of an example signal (**left**) taken from a dataset of real respiration signals [254] (RR of 20 bpm −0.33 Hz-, and sampling frequency of 50 Hz). (**B**) Multiresolution analysis (MRA) decomposition process (**top**). The lower part shows an example of the MRA analysis applied to the signal above ((**A**), **left**). Six-level decomposition was applied using the ‘Haar’ wavelet. Two detail levels and the sixth approximation level are represented. The spectrum of the approximation coefficients (level 6) was obtained.

**Figure 33 sensors-20-05446-f033:**
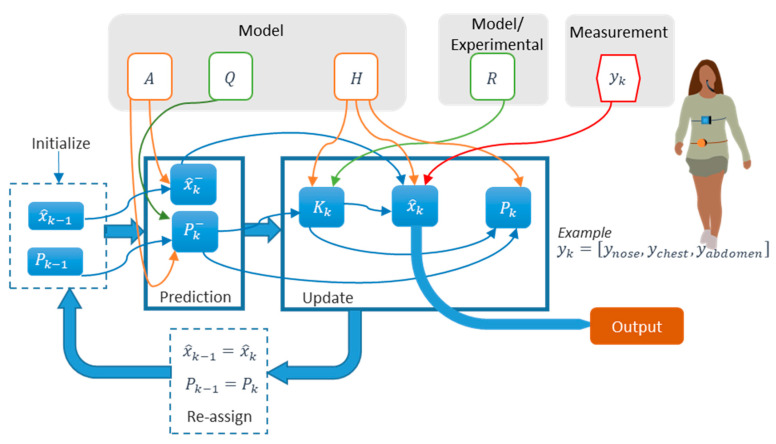
Kalman filter algorithm for the fusion of different respiration sensors.

**Figure 34 sensors-20-05446-f034:**
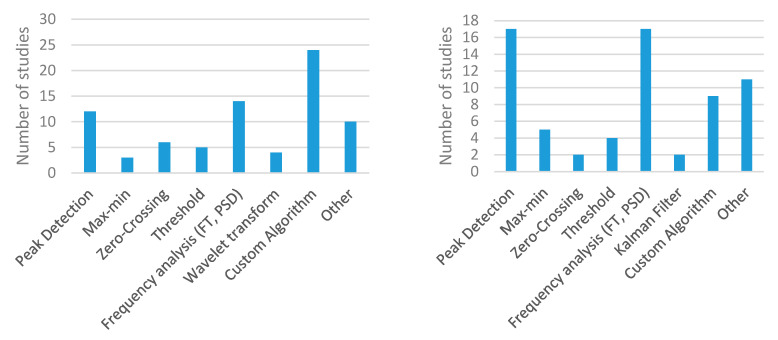
Number of studies using different processing algorithms for the wearable (**left**) and environmental (**right**) categories.

**Figure 35 sensors-20-05446-f035:**
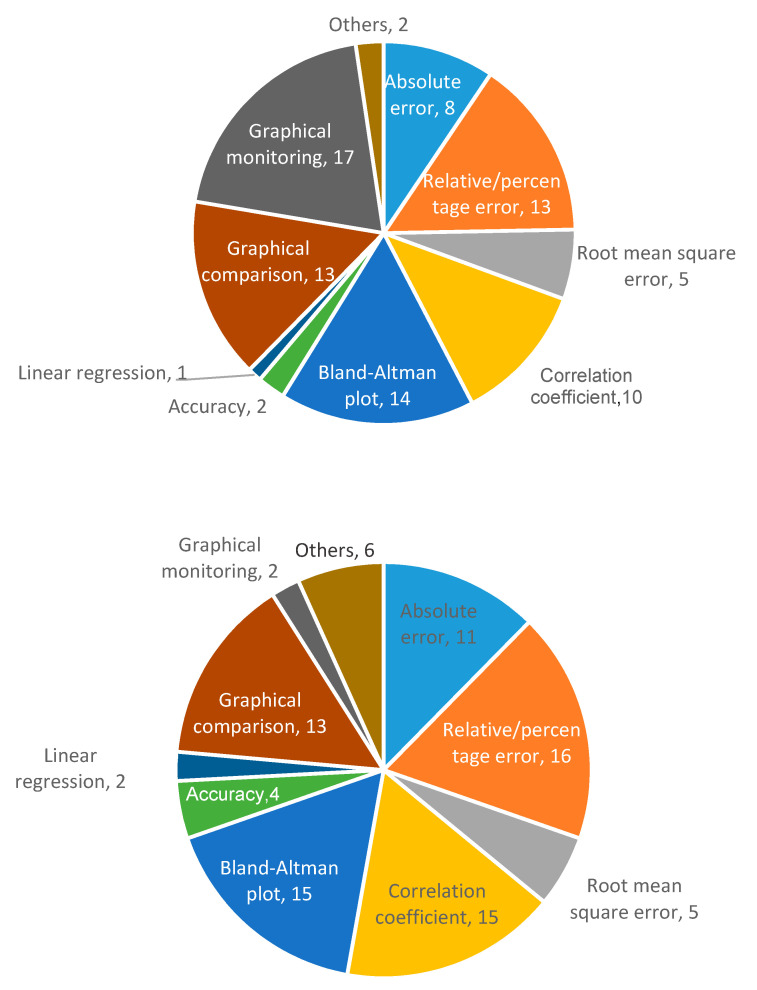
Number of studies using the different figures of merit to determine sensor performance for the wearable (**top**) and environmental (**bottom**) categories.

**Figure 36 sensors-20-05446-f036:**
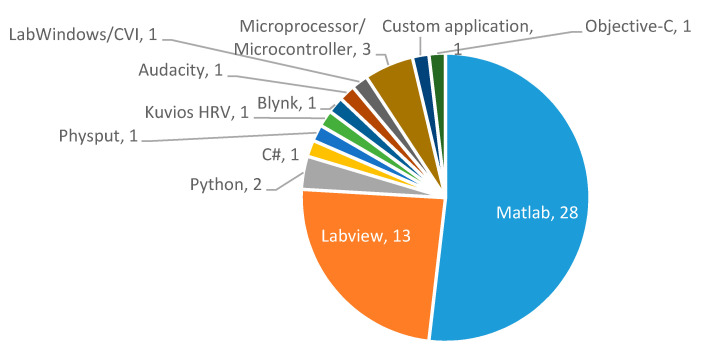
Number of studies using the different processing tools for the wearable (**top**) and environmental (**bottom**) categories.

**Table 1 sensors-20-05446-t001:** Analysis of techniques, sensors, breathing parameters, and sensor locations and sizes for studies of the wearable category.

Study ^1^	Technique	Sensor	Measured Parameter	Location	Size
Aitkulov 2019 [57,58]	Chest wall movements	Fiber optic	RR	Chest	-
Balasubramaniyam 2019 [59]	Chest wall movements	Resistive	RR	Abdomen (shirt)	-
Bricout 2019 [60]	Chest wall movements	Accelerometer	RR	Chest Abdomen	-
Chu 2019 [61]	Chest wall movements	Resistive	RR TV	Chest	
Elfaramawy 2019 [62]	Chest wall/abdomen movements	Accelerometer Gyroscope Magnetometer	RR	Chest Abdomen	26.67 × 65.53 mm
Fajkus 2019 [63]	Respiratory air flow	Fiber optic	RR	Nose (nasal oxygen cannula)	
Hurtado 2019 [64]	Air temperature	Pyroelectric	RR	Nose (below)	30 × 16 × 20 mm
Jayarathna 2019 [65]	Chest wall movements	Resistive	RR	Chest (shirt)	-
Kano 2019 [66]	Air humidity	Nanocrystal and nanoparticles	RR	Mouth mask	
Karacocuk 2019 [67]	Chest wall movements	Accelerometer Gyroscope	MV	Chest (front and back)	
Massaroni 2019 [68]	Respiratory air flow (pressure)	Differential pressure	RR	Nose (nostril)	36 mm diameter (PCB)
Massaroni 2019 [69]	Chest wall movements	Resistive	RR	Chest and abdomen (shirt, front and back)	-
Nguyen 2019 [70]	Respiratory air flow (vibration)	Differential pressure sensor	RR	Nose (nasal bridge)	
Presti 2019 [71]	Respiratory air flow	Fiber optic	RR	Cervical spine	90 × 24 × 1 mm
Presti 2019 [72]	Chest/abdomen movements	Fiber optic	RR	Chest	-
Puranik 2019 [73]	Chest wall movements	Gyroscope	Monitoring of breathing	Chest Abdomen	-
Soomro 2019 [74]	Air humidity	Impedance	Monitoring of breathing	Nose (below)	
Xiao 2019 [75]	Air humidity	Resistive	Monitoring of breathing	Mouth mask (2–3 cm from nose)	
Yuasa 2019 [76]	Respiratory sounds Chest wall movements	MicrophoneOptical	RR	Chest (adhesive gel)	
Zhang 2019 [77]	Chest wall movements	Triboelectric nanogenerator	RR	Abdomen	-
Dan 2018 [78]	Chest wall movements	Accelerometer	RR Respiratory phase	Neck (Suprasternal notch area)	-
Koyama 2018 [79]	Chest wall movements	Fiber Optic sensor	RR	Abdomen (Cardigan, garment)	-
Malik 2018 [80]	Air humidity	Capacitive sensor	Monitoring of breathing	Mouth mask	-
Martin 2018 [81]	Respiratory sounds	Microphone	RR	Head (inside ear)	-
Pang 2018 [82]	Air humidity	Nanocrystal and Nanoparticles sensor	Monitoring of breathing	Mouth mask	-

^1^ Note: The analysis for studies published before 2018 [2,3,17,21,49,83,84,85,86,87,88,89,90,91,92,93,94,95,96,97,98,99,100,101,102,103,104,105,106,107,108,109,110,111,112,113,114,115,116,117,118,119,120,121,122,123,124,125,126,127,128,129,130,131,132,133,134,135,136,137,138,139,140,141,142,143,144,145,146,147,148,149,150,151,152,153,154,155,156,157,158,159,160,161,162] is included in Appendix A (Table A1).

**Table 2 sensors-20-05446-t002:** Analysis of sensing techniques, sensors, breathing parameters, and sensor location and size for studies of the environmental category.

Study ^1^	Technique	Sensor	Measured Parameter	Location	Size
Al-Wahedi 2019 [163]	Modulation cardiac activity	Radar	RR	Distance from subject (20–75 cm away)	
Chen 2019 [164]	Modulation cardiac activity	Radar	RR	Mat (below bed)	
Gunaratne 2019 [165]	Chest wall movements	Piezoelectric	RR	Mat	7 × 7 cm (each sensor)
Guo 2019 [166]	Chest wall movements	Capacitive	RR	Mat	
Isono 2019 [167]	Chest wall movements	Piezoelectric	RR	Others (under bed legs)	
Ivanovs 2019 [168]	Chest wall movements Modulation cardiac activity	Camera Radar	Respiration detection	-	
Joshi 2019 [169]	Chest wall movements	Capacitive	RR	Mat (below baby mattress)	580 × 300 × 0.4 mm
Krej 2019 [170]	Chest wall movements	Fiber optic	RR	Mat	
Lorato 2019 [171]	Air temperature	Camera	RR	Distance from subject (side and front, 10–50 cm away)	
Massaroni 2019 [172]	Chest wall movements	Camera	RR	Distance from subject (1.2 m away)	
Park 2019 [173]	Chest wall movements	Piezoelectric	RR	User’s mat (Chest region)	40 × 750 × 0.25 mm
Walterscheid 2019 [174]	Modulation cardiac activity	Radar	RR	Distance from subject (3.3–4.2 m away)	
Wang 2019 [175]	Modulation cardiac activity	Radar	RR	Distance from subject (50 cm away)	
Xu 2019 [176]	Respiratory sounds	Microphone	RR	Others (instrument panel of vehicle)	
Yang 2019 [177]	Modulation cardiac activity	Radar	RR	Distance from subject (1.5 m height, 0–3 m away)	
Chen 2018 [178]	Modulation cardiac activity	Wi-Fi transmitter and receiver	RR Respiration detection	Nodes	-
Chen 2018 [179]	Chest wall movements	Piezoelectric	RR	Mat	2 × 35 cm
Massaroni 2018 [180]	Chest wall movements	Camera	RR Respiratory pattern	Distance from subject (1.2 m away)	-
Massaroni 2018 [181]	Chest wall movements	Fiber optic	RR	Others (inside ventilator duct)	3 cm
Sadek 2018 [182]	Chest wall movements	Fiber optic	RR Respiratory pattern	Mat	20 × 50 cm

^1^ Note: The analysis for studies published before 2018 [5,6,7,9,10,19,48,50,51,52,53,54,183,184,185,186,187,188,189,190,191,192,193,194,195,196,197,198,199,200,201,202,203,204,205,206,207,208,209,210,211,212,213,214,215,216,217,218,219,220,221,222,223,224,225,226,227,228,229,230,231,232,233,234] is included in Appendix A (Table A2).

**Table 3 sensors-20-05446-t003:** Comparison of the main transmission technologies used in respiratory monitoring systems [237].

	Operating Bandwidth	Transmission Speed	Power Consumption	Range (m)	Hardware Complexity
Bluetooth	2.4 GHz	1 Mbps	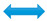	1–100	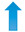
Zigbee	2.4 GHz (valid worldwide)	250 kbps at 2.4 GHz band	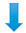	10–100	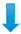
Wi-Fi	2.4–5 GHz generally	Up to 1 Gbps	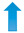	50–100	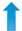
GSM/GPRS	850–1900 MHz	120 kbps	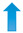	100 m–several kilometers	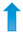
Radio frequency	433 MHz	4 kbps	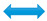	20–200	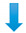

**Table 4 sensors-20-05446-t004:** Analysis of transmission technology, processing station, and energy autonomy for studies in the wearable category.

Study ^1^	Wireless Transmission	Wired Transmission	Processing Station	Battery Capacity	Battery Life (Type Battery)
Aitkulov 2019 [57,58]	-	Data storage	-	-	-
Balasubramaniyam 2019 [59]	Internet connection	-	Cloud storage, PC, Smartphone	-	-
Bricout 2019 [60]	-	-	-	-	-
Chu 2019 [61]	Bluetooth	-	PC	-	-
Elfaramawy 2019 [62]	Radio-frequency	-	PC	3.7 V, 100 mAh	6 h (Li-ion battery)
Fajkus 2019 [63]	-	Interrogator DAQ (data acquisition)	PC	-	-
Hurtado 2019 [64]	-	-	-	-	-
Jayarathna 2019 [65]	Bluetooth (low energy), SD card	-	PC, smartphone, cloud Storage	600 mAh	5 days (Li-ion battery)
Kano 2019 [66]	Bluetooth	-	Smartphone	3 V	(Cell battery)
Karacocuk 2019 [67]	Bluetooth	-	PC, smartphone	-	-
Massaroni 2019 [68]	Bluetooth	-	PC	3.6 V, 650 mAh	8 h (Li-polymer battery)
Massaroni 2019 [69]	Bluetooth	-	-	-	-
Nguyen 2019 [70]	-	-	-	-	-
Presti 2019 [71]	-	Interrogator	PC	-	-
Presti 2019 [72]	-	Interrogator	PC	-	-
Puranik 2019 [73]	Wi-Fi	-	-	3.7 V, 1020 mAh	(Li-ion battery)
Soomro 2019 [74]	-	USB	PC, smartphone	-	-
Xiao 2019 [75]	-	-	PC	-	-
Yuasa 2019 [76]	-	USB	Smartphone	-	-
Zhang 2019 [77]	-	-	Smartphone, PC	-	-
Dan 2018 [78]	-	-	-	-	-
Koyama 2018 [79]	-	Interrogator DAQ	PC	-	-
Malik 2018 [80]	-	DAQ	-	-	-
Martin 2018 [81]	-	-	PC	-	-
Pang 2018 [82]	-	-	-	-	-

^1^ Note: The analysis for studies published before 2018 [2,3,17,21,49,83,84,85,86,87,88,89,90,91,92,93,94,95,96,97,98,99,100,101,102,103,104,105,106,107,108,109,110,111,112,113,114,115,116,117,118,119,120,121,122,123,124,125,126,127,128,129,130,131,132,133,134,135,136,137,138,139,140,141,142,143,144,145,146,147,148,149,150,151,152,153,154,155,156,157,158,159,160,161,162] is included in Appendix A (Table A3).

**Table 5 sensors-20-05446-t005:** Analysis of validation experiments for the studies in the wearable and environmental categories.

Validation Parameters	Number of Subjects						Duration		Activities Considered					
	1	2 to 5	6 to 10	11 to 20	>20	Not specified	<5 min	>5 min	Sitting	Standing	Lying down	Sleeping	Walking, running, moving	Motion artifacts
**Number of studies**	34	40	30	22	19	63	47	34	59	25	66	16	28	27

**Table 6 sensors-20-05446-t006:** Analysis of the processing algorithm, performance evaluation, and software for the studies of the wearable category.

Study ^1^	Algorithm	Performance Evaluation	Performance Value	Analysis Software
Aitkulov 2019 [57,58]	Frequency analysis	Graphical comparison	-	-
Balasubramaniyam 2019 [59]	-	-	-	MATLAB
Bricout 2019 [60]	Adaptive reconstruction	Correlation factor	0.64–0.74	-
Chu 2019 [61]	Peak detection	Bland-Altman analysis Correlation factor	0.99 (correlation)	MATLAB
Elfaramawy 2019 [62]	Peak detection	-	-	MATLAB
Fajkus 2019 [63]	Peak detection	Relative error Bland-Altman analysis	3.9% (RE)	Labview
Hurtado 2019 [64]	Zero-crossing detection	Relative error Bland-Altman analysis	0.4 bpm (BA, mean of difference –MOD–)	-
Jayarathna 2019 [65]	Peak detection	-	-	-
Kano 2019 [66]	Peak detection	Correlation coefficient Bland-Altman analysis	0.88 (correlation) 0.026 bpm (BA, MOD)	-
Karacocuk 2019 [67]	Frequency analysis	Correlation	-	MATLAB Microprocessor
Massaroni 2019 [68]	Custom algorithm	Relative error Linear regression Bland-Altman analysis	4.03% (RE) 0.91–0.97 (*r*^2^) −0.06 (BA, MOD)	MATLAB
Massaroni 2019 [69]	Peak detection	Bland-Altman analysis	0.05 bpm (BA, MOD)	-
Nguyen 2019 [70]	Frequency analysis	-	-	-
Presti 2019 [71]	Peak detection	Percentage error	<4.71% (PE)	MATLAB Labview
Presti 2019 [72]	Peak detection	-	-	-
Puranik 2019 [73]	-	-	-	-
Soomro 2019 [74]	-	-	-	-
Xiao 2019 [75]	-	Graphical comparison	-	-
Yuasa 2019 [76]	Peak detection	Accuracy	61.3–65.6%	MATLAB
Zhang 2019 [77]	Frequency analysis	-	-	-
Dan 2018 [78]	Zero-crossing detection	Bland-Altman analysis	0.01–0.02 bpm (BA, MOD)	-
Koyama 2018 [79]	Frequency analysis	Absolute error	4 bpm	Python
Malik 2018 [80]	-	Graphical monitoring	-	Python
Martin 2018 [81]	Custom algorithm	Mean absolute error Mean relative error Bland-Altman analysis	2.7 bpm (MAE) 30.9% (MRE) 2.4 bpm (BA, MOD)	MATLAB
Pang 2018 [82]	-	Graphical monitoring	-	-

^1^ Note: The analysis for studies published before 2018 [2,3,17,21,49,83,84,85,86,87,88,89,90,91,92,93,94,95,96,97,98,99,100,101,102,103,104,105,106,107,108,109,110,111,112,113,114,115,116,117,118,119,120,121,122,123,124,125,126,127,128,129,130,131,132,133,134,135,136,137,138,139,140,141,142,143,144,145,146,147,148,149,150,151,152,153,154,155,156,157,158,159,160,161,162] is included in Appendix A (Table A4).

**Table 7 sensors-20-05446-t007:** Analysis of the processing algorithm, performance evaluation, and software for the studies of the environmental category.

Study ^1^	Algorithm	Performance Evaluation	Performance Value	Analysis Software
Al-Wahedi 2019 [163]	Frequency analysis	Manual verification Relative error	4–14% (RE)	Labview
Chen 2019 [164]	Zero-crossing detection	Mean squared error	1.23 bpm	-
Gunaratne 2019 [165]	Wavelet transform Fuzzy logic	Relative error	6.2%	-
Guo 2019 [166]	Wavelet transform	Cross-correlation	0.76–0.85	-
Isono 2019 [167]	Custom algorithm	Linear regression Bland-Altman analysis	0.969 (*r*^2^) 0.07–0.17 bpm (BA, MOD)	LabChart MATLAB
Ivanovs 2019 [168]	Neural networks	Others	-	-
Joshi 2019 [169]	-	Correlation factor Root mean square error Bland-Altman analysis	0.74 (correlation) 4.7 bpm (RMSE) −0.36 (BA, MOD)	-
Krej 2019 [170]	Machine learning methods	Root mean square error Bland-Altman analysis	1.48 bpm (RMSE) 0.16 bpm (BA, MOD)	C# R
Lorato 2019 [171]	Frequency analysis	Root mean square error Bland-Altman analysis	1.59 bpm (RMSE)	MATLAB
Massaroni 2019 [172]	Custom algorithm	Absolute error Standard error Percentage error Bland-Altman analysis	0.39 bpm (AE) 0.02 bpm (SE) 0.07% (PE) −0.01 bpm (BA, MOD)	MATLAB
Park 2019 [173]	Frequency analysis	Accuracy Bland-Altman analysis	99.4% (Acc)	MATLAB
Walterscheid 2019 [174]	Peak detection	Graphical comparison	-	-
Wang 2019 [175]	Custom algorithm	Absolute error Relative error	0.3 bpm (AE) 2% (RE)	MATLAB
Xu 2019 [176]	Custom algorithm	Absolute error Correlation factor	0.11 bpm (AE) 0.95 (correlation)	-
Yang 2019 [177]	Custom algorithm	Absolute error	0.3–0.6 bpm	-
Chen 2018 [178]	Custom algorithm	Accuracy	98.65%	-
Chen 2018 [179]	Frequency analysis	Graphical comparison	-	Mobile app
Massaroni 2018 [180]	Threshold detection Zero-crossing detection Custom algorithm	Correlation factor Bland-Altman analysis Percentage error Others	0.97 (correlation) 0.01 bpm (BA, MOD) 5.5% (PE)	MATLAB
Massaroni 2018 [181]	Peak detection	Relative error	2%	MATLAB
Sadek 2018 [182]	Peak detection Custom algorithm	Correlation factor Bland-Altman analysis Mean absolute error	0.78 (correlation) 0.38 bpm (MAE)	-

^1^ Note: The analysis for studies published before 2018 [5,6,7,9,10,19,48,50,51,52,53,54,183,184,185,186,187,188,189,190,191,192,193,194,195,196,197,198,199,200,201,202,203,204,205,206,207,208,209,210,211,212,213,214,215,216,217,218,219,220,221,222,223,224,225,226,227,228,229,230,231,232,233,234] is included in Appendix A (Table A5).

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
