# Peer review of "Sensing Systems for Respiration Monitoring: A Technical Systematic Review"

_sensors, 2020, doi:10.3390/s20185446_

Round 1
Reviewer 1 Report
Report on: Sensing systems for respiration monitoring: a systematic review
by Erik Vanegas et al.
The authors present a systematic review focused on techniques for respiratory monitoring. It is very interesting and well written. Although there are some reviews on this topic and the number of studies is increasing in recent years, the authors presented a review with some novelties with respect to the ones reported in the literature and they clearly highlighted these novelties. In addition and as said before, studies, the development of new systems, and the interest of companies on respiratory monitoring are increasing fast, so a review on this topic is absolutely welcome.
Summing up, the article is clearly written, well organized, and in absolutely topic with this high-impacted journal and with the proposed special issue, so I hope it will be published. However, I have the following comments:
-) Introduction (Page 1). “..and may server…” did you mean “..and may serve..”?
-) Introduction (page 2). The authors said: “..Monitoring systems use sensors to measure breathing parameters. There are large differences among approaches depending on…” In the list after “depending on” I think the authors should delate “validation method” because I think that the validation method does not influence the approach used for respiratory monitoring (maybe the application field) and think the authors should replace “experiments carried out to test the sensors” with “field of application”.
-) M & M (page 5). The authors said, “For each search, the 100 most relevant papers were selected.” How did you choose the most relevant? For instance, I found that some recent articles on respiratory monitoring are not included in this review. For instance:
“Lo Presti D. et al. (2019). Cardio-respiratory monitoring in archery using a smart textile-based on flexible fiber Bragg grating sensors. Sensors, 19(16), 3581”
“Massaroni, C., et al. "Fiber Bragg Grating Sensors for Cardiorespiratory Monitoring: A Review." IEEE Sensors Journal (2020)”
-) Results (page 7). “According to the review of Masseroni” please replace Masseroni with Massaroni (also in other parts of the article)
-) Results (page 7). “..Differential flowmeters (pressure sensors)..” it seems that differential flowmeters are pressure sensors. Please replace it with “differential flowmeters”.
-) Results (page 7). Please define all the acronyms (e.g., DC generators, ECG, PPG…)
-) Results (page 9). Figure 4 and some breathing parameters reported in the figure are not clear. For instance, what is the signal reported in fig 4.A? I think it should be important to clarify if it is related to respiratory volume or to respiratory flow. In addition, the window used to highlight inspiratory and expiratory time are different from the interval ta-tb and tc-td. This representation should be misleading.
-) Results (page 9) “The location of a sensor largely depends on its operating principle.” I think it should be better to highlight that is also important the breathing parameters that the system wants to monitor and the application. You may change the sentence above with “The location of a sensor largely depends on its operating principle and the specific application”
-) Results (page 13). Figure 6, please replace “air temperat.” with “air temperature”.
-) Results (page 18). Please use the correct units of measurements. Replace MHZ with MHz.
-) Results (page 27). Table 4. In some cases, the use of wired and wireless transmission is not specified because this information is expected to consider the specific set-up. For instance, the interrogators used in [68] and [69] use a wired communication, in addition, the connection between the sensors and the interrogator is needed (fiber Bragg grating sensors must be connected to the interrogator. Please improves this part that involves table 4 and figure 20.
-) the bar graphs in figures 6, 7, 8, 9, 10, 21, 22 are useful, but I recommend using in all cases either the number of studies or the percentage. Now in some cases, authors used the number of studies (e.g., figures 6, 7, 8,…, 28) and in other cases the percentage (e.g., figures 21 and 22). A further solution can be to build a double-axis graph in all cases (the two axes are the percentage and the number of studies): This observation may be also considered for the pie chart.
-) Results (page 39). I do not think it is useful to report a figure dedicated to explaining the Bland Altman plot. The aim of this review is much different. So, please remove figure 29.
-) Discussion (page 54-57). The discussion section must be improved.
For instance, lines 1319-1349. The authors shortly describe the two sets of systems for respiratory monitoring. But they should highlight other aspects, for instance regarding wearable they did not describe concerns related to artifacts caused by respiratory unrelated movements.
In addition, the authors highlighted the challenges related to sensor aging and long-term behavior. But in some cases, this is not a significant concern since the sensing part can be replaced due to the low cast. Please comment on this aspect.
Again, the authors said “If the purpose of the study is to validate a sensor by comparing its performance with a reference system, the Bland-Altman analysis is the most suitable instrument. ” I do not agree with this statement. There are different methods in metrology to describe the performance of a system and I think Bland-Altman analysis is just one of them but not the best one. Why did you think that the B&A analysis is the most suitable?
Again (lines 1394-1400), I agree that the most common parameters obtained in the studies are the respiratory rare, and I think this is just because it is very complex to obtain an accurate estimation of respiratory volumes (tidal volume, minute volume…) with some techniques (in particular, almost all the wearable systems). May you add a sentence that makes this message strong?
Finally, in section results, the authors reported an interesting analysis regarding the number of studies (or their percentage) of wearable category (figure 6, 7…). I expect comments in the discussion section. Basically, I expect comments which answer several questions. For example, considering figure 6, the questions may be: “why are fiber optics the most used sensors for wearable systems? And more in general, an explanation about why there is that distribution (bar graphs in fig 6) regarding the use of sensors for developing wearable systems for respiratory monitoring. I just made the example of figure 6, but I expect similar comments for the data reported in Figures 7, 8, 9, 10…
Reviewer 2 Report
The manuscript provides the results of a very ambitious attempt to review solutions used for respiratory monitoring. Overall, the reader will appreciate the scope and detail of the analysis. However, there are some aspects of the methodological approach, which would require an explanation.
The title of the manuscript is very broad, and it does not fully reflect the content. It seems that paper is about the design of the sensing systems used for respiratory monitoring. Maybe it is worth to explain from the beginning that it is technical and not clinical report.
The approach to searching strategy is not entirely clear. As the search items are quite broad, one can imagine that many papers after their use could have been retrieved. In one of the charts, the authors tend to claim that they analysed “100 most relevant papers per search”. Such a formulation of the search strategy is not fully understandable. What does “the most relevant” mean? How was the decision made that one paper is more relevant than another? The inclusion and exclusion criteria must be specified unequivocally.
The search strategy and tackling with retrieved records is not explained transparently either.
In the case of the systematic review, the PRISMA diagram is usually provided to reflect the process of the items selection. It lacks in this manuscript.
Round 2
Reviewer 1 Report
After this first round of review, the article can be accepted